# The Leaderboard Illusion

**Shivalika Singh**[1]* **Yiyang Nan**[1] **Alex Wang**[2] **Daniel D'souza**[1] **Sayash Kapoor**[3]

**Ahmet Üstün**[1] **Sanmi Koyejo**[4] **Yuntian Deng**[5] **Shayne Longpre**[6]

**Noah A. Smith**[7,8] **Beyza Ermis**[1] **Marzieh Fadaee**[1]

**Sara Hooker**[1]

[1]Cohere Labs   [2]Cohere
[3]Princeton University   [4]Stanford University
[5]University of Waterloo   [6]Massachusetts Institute of Technology
[7]Allen Institute for Artificial Intelligence   [8]University of Washington

## Abstract

Measuring progress is fundamental to the advancement of any scientific field. As benchmarks play an increasingly central role, they also become more susceptible to distortion. Chatbot Arena has emerged as the go-to leaderboard for ranking the most capable AI systems. Yet, in this work we identify systematic issues that have skewed the competitive landscape. Specifically, undisclosed private testing practices benefit a handful of providers who are able to test multiple variants before public release and selectively retract scores. We establish that the ability of these providers to choose the best score leads to biased Arena scores due to selective performance disclosure. At an extreme, we found one provider testing 27 private variants before making one model public at the second position on the leaderboard. We also show that proprietary closed models are sampled at higher rates (i.e., involved in more battles) and are less likely to removed from the arena compared to open-weight and open-source models. These policies lead to large data access asymmetries over time. The top two providers have individually received an estimated 19.2% and 20.4% of all data on the arena, while 83 open-weight models collectively received only 29.7%. Even conservative estimates indicate that access to Chatbot Arena data offers substantial benefits: limited additional data can boost relative performance by up to 112% on ArenaHard, a test set from the arena distribution. These dynamics lead to overfitting on Arena-specific dynamics rather than reflecting general model quality. The Arena builds on the substantial efforts of both the organizers and an open community that maintains this valuable evaluation platform. We offer actionable recommendations to make Chatbot Arena's evaluation framework fairer and more transparent for the field.

## 1 Introduction

Benchmarks have long played an integral role in the development of machine learning systems, serving as pivotal instruments for evaluating progress, guiding research agendas, and influencing funding decisions [13, 42]. Over time, this impact ranges from early shared tasks in information retrieval and machine translation [34, 43], through large-scale image classification benchmarks such as ImageNet [21], to contemporary evaluation frameworks like GLUE that catalyzed advances in

39th Conference on Neural Information Processing Systems (NeurIPS 2025) Track on Datasets and Benchmarks.

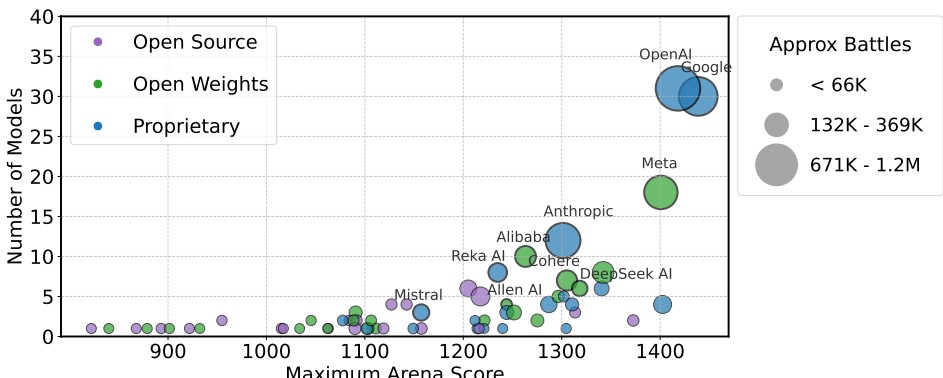

Figure 1: **Number of public models vs. maximum arena score per provider.** Marker size indicates total number of battles played. Proprietary providers typically achieve higher scores, correlating with both the number of models released and the number of Arena battles played. As discussed in Section 3 and Section 4, this increased exposure may yield advantages via model selection and adaptation to the Arena distribution. Figure reflects publicly disclosed results as of April 23rd, 2025.

natural language understanding [77]. Recently, the rapid ascent and broad adoption of generative AI systems, accompanied by significant increases in computational resources and public attention [41, 36, 67, 37], have intensified both the importance and critical examination of leaderboards used to assess model capabilities [59].

Established in 2023, *Chatbot Arena* quickly became the predominant benchmark for comparing LLMs. Unlike traditional static benchmarks, Chatbot Arena is dynamic and user-driven, allowing unrestricted prompts and daily updates to better reflect real-world scenarios that traditional evaluations miss [74, 11, 58]. Its widespread adoption across industry, academia, and media highlights its significant influence on perceptions of model quality and technical progress.

Despite these advantages, heavy reliance on a single leaderboard inherently risks distorting the objectives it was intended to measure, exemplifying Goodhart's Law, where *"when a measure becomes a target, it ceases to be a good measure"* [26, 73]. Motivated by this concern, we perform a systematic empirical study of Chatbot Arena by analyzing approximately 2M battles and 243 models from 42 distinct providers over a 16-month period (January 2024–April 2025). Our investigation uncovers several critical limitations affecting the reliability and fairness of the Chatbot Arena benchmark:

**1. Preferential treatment around private testing and retraction.** Chatbot Arena permits select providers to test numerous private model variants in parallel – up to 27 in a single month – without requiring public release. There is no guarantee that leaderboard entries match publicly accessible APIs[1]. Simulations and real-world trials show that choosing the top performer from $N$ submissions can significantly inflate ratings, enabling systematic leaderboard gaming.

**2. Far more data is released to proprietary model providers.** Chatbot Arena relies on community-generated feedback, yet evaluation data is distributed unequally. Some proprietary models have received as much as 20.4% of all test prompts, while a combined 41 fully open-source models received only 8.9% of the total. These estimates are derived from the share of total battles played by each provider's models, as shown in Figure 4. The resulting data imbalance significantly disadvantages open-source and open-weight models in adapting to in-distribution prompts.

**3. Chatbot Arena data access drives significant performance gains.** We find that access to Chatbot Arena data materially improves model performance on the leaderboard. In a controlled experiment, increasing training exposure to Chatbot Arena data from 0% to 70% more than doubled win rates on the ArenaHard benchmark [46] – from 23.5% to 49.9%. This likely underestimates the total effect, as some providers also have access to private API data unavailable to others, which could further amplify performance gains.

**4. Deprecations can result in unreliable model rankings.** Of the 243 public models we tracked, 205 were silently removed from the leaderboard – far more than the 47 officially deprecated in

---

[1]https://www.theverge.com/meta/645012/meta-llama-4-maverick-benchmarks-gaming

Chatbot Arena's backend (FastChat)[2]. These removals violate assumptions of the Bradley-Terry model [8], undermining rating stability. Notably, 64% of silently deprecated models are open-weight or open-source, suggesting uneven impact.

This paper provides a detailed, data-driven critique of Chatbot Arena, revealing systematic biases and transparency gaps that undermine its reliability as a benchmark. We conclude with concrete, actionable recommendations to improve fairness and ensure the leaderboard remains a credible tool for evaluating and advancing generative AI systems.

## 2    Overview of Methodology

This study draws on four complementary data sources spanning January 2024-April 2025, covering approximately 2M Chatbot Arena battles and 243 distinct models. These include: (1) public battle datasets released by Chatbot Arena, (2) a three-month scrape of live Arena battles to identify private variants and sampling patterns.(3) provider-side API logs from models controlled by the authors, and (4) leaderboard snapshots tracking model scores and status. We describe our data sources in detail in Appendix E.

We analyze model rankings through the **Bradley-Terry (BT) model**, the foundation of the Arena Score. In Chatbot Arena, users are presented with two anonymous model responses and vote for the preferred one (or select a tie). The BT model uses these pairwise outcomes to infer each model's latent skill level. Specifically, if model $i$ beats model $j$ in a pairwise comparison, the BT model adjusts their respective scores such that the probability of $i$ beating $j$ is proportional to the ratio of their skill parameters. The Arena Score is a normalized transformation of these latent skill parameters. It reflects not only a model's overall win-rate, but also the strength of the opponents it defeats. Unlike simpler averaging methods, the BT formulation accounts for ties and missing matchups, making it more robust to sparsity and incremental model additions.

However, the reliability of Arena Scores hinges on key assumptions: unbiased sampling, consistent evaluation conditions, and sufficient connectivity among models via shared opponents. We investigate how violations of these assumptions – such as private variant retractions, uneven sampling, or extensive deprecations – can distort rankings and reduce the trustworthiness of Arena Scores. Our methodology combines the above data sources to quantify the prevalence and effects of these issues, supporting our analysis in subsequent sections.

## 3    Results: Impact of Private Testing and Selective Retraction on Arena Scores

### 3.1    Preferred Providers Frequently Use Private Testing

Although not an officially stated policy[3], our audit of Chatbot Arena data using `scraped-random-sample` revealed that providers are permitted to test multiple private model variants simultaneously, without any obligation to publicly release or de-anonymize these submissions. In Figure 9 (see Appendix H.4), we show the number of private variants we tracked as belonging to each provider from January to March 2025. Meta and Google had the most active private models during this period, with 27 and 10 tracked models, respectively. Meta's peak of private testing closely preceded the release of its Llama 4 models [56], while Google's tests were mostly associated with its proprietary Gemini models, with only a single observed instance involving the open-weight Gemma 3 [33].

We note this is likely a conservative estimate as it only tracks the private variants on the main Chatbot Arena, and does not take into account private variants on specialized leaderboards run by Arena such as for vision or code. For example, when we include Meta's private models from the vision leaderboard, we identify an additional 16 variants, bringing its total to 43. In contrast, smaller startups, such as Reka, were found to have one active private variant live in the arena. Notably, we found that no private models were tested by academic labs during the observed period. This disparity suggests that only certain providers may have been aware they could submit multiple private variants, as we observe clear differences in the number and frequency of private testing among providers.

---

[2]`http://github.com/lm-sys/FastChat/`
[3]`https://drive.google.com/file/d/1reook2cjwq81xD6Yn528KOLWeWRyOZvN/view?usp=sharing`

## 3.2 Simulated Experiments on Private Testing and Retraction

Private testing coupled with the option to retract enables a best-of-N strategy, where an organization submits multiple model variants to Chatbot Arena, privately evaluates them, and retains only the top-performing variant to be publicly published on the leaderboard. In this section, we show that best-of-N submissions violate the BT unbiased sampling assumption. This systematically inflates model rankings and distorts the leaderboard ranking.

**Unbiased Sampling Assumption.** To study the selection bias scenario, assume a provider submits $N$ variants of a model, each variant $k$ having a true underlying skill parameter $\beta_k$, sampled from a distribution centered at some base skill level $\beta$. Each variant's observed skill is estimated using $\hat{\beta}_k$ where $\hat{\beta}_k$ explicitly serves as an estimator for the true parameter $\beta$. The probability of observing an exceptionally high-performing variant increases with the number of submissions $N$. Thus, the observed skill of the submitted model is: $\hat{\beta}_{\text{Best}} = \max\{\hat{\beta}_1, \hat{\beta}_2, \dots, \hat{\beta}_N\}$.

Since each $\hat{\beta}_k$ is subject to statistical fluctuation due to finite match sampling, selecting the best variant based on observed performance introduces an upward bias. Specifically, the expected value of the best-performing variant is *strictly greater than* that of a regular submission:

$$\mathbb{E}[\hat{\beta}_{\text{Best}}] > \mathbb{E}[\hat{\beta}_k], \qquad \forall k \in \{1, 2, \dots, N\}. \tag{1}$$

where the draws are *non-degenerate* ($\text{Var}(\hat{\beta}_k) > 0$) and $N \geq 2$ (see Appendix D for further details). This violates the BT model's assumption of unbiased sampling and alters the likelihood landscape. The reported rating no longer reflects a single, unbiased estimate of skill, but an extreme value from multiple independent estimations. As a result, the BT estimator systematically inflates the ratings of models submitted under the best-of-N strategy, distorting leaderboard rankings.

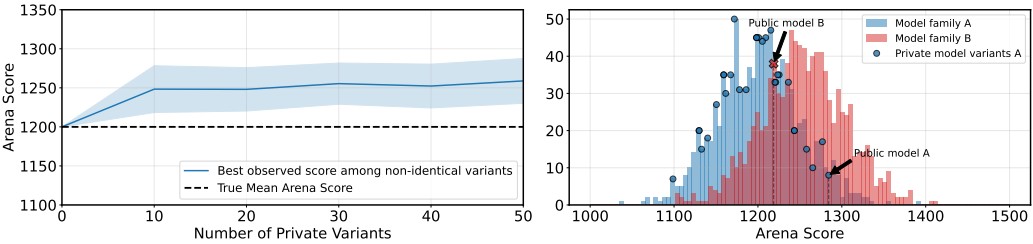

Figure 2: **Left: Impact of the number of private variants tested on the best Expected Arena Score.** As more private variants are tested—and their Arena Scores revealed—the likelihood of selecting a model from the upper end of the performance distribution increases, allowing the provider to effectively identify the highest-scoring variant. **Right: Simulated impact of best-of-N submission strategies on Arena leaderboard rankings. Model family A** has a lower average Arena Score than **Model family B**, yet by submitting multiple private variants and selecting the best-performing one, it can surpass the only public submission from **Model family B**.

**Role of the number of private variants.** In Chatbot Arena, we observe an asymmetry in the number of private models tested, with one provider testing up to 27 variants before launch. To investigate the impact, we simulated the expected lift in Arena Score as the number of private variants increased from 0 to 50. Each candidate $k$ was assigned a *true* Arena score: $E_k \sim \mathcal{N}(\mu = 1{,}200, \ \sigma_k = 25)$, and evaluated with $n = 3000$ synthetic votes using the BT model. After selecting the variant with the highest estimated Arena score, we found a significant extreme-value uplift: testing $M = 20$ private variants increased the expected maximum score by approximately 50 Arena score compared to a single public submission, while $M = 50$ pushed the advantage around 70 Arena score. (see Figure 2, left). This highlights the advantage gained by providers who test multiple private variants. The 1200 baseline (shown in Figure 2, left) serves as a reference point representing the expected performance of a single randomly submitted checkpoint in our experimental framework. We provide more details about this simulation in Appendix M.

**Asymmetries in which providers have access to private testing.** We observe in practice that only a few preferred providers were able to test many variants and handpick the best result. As we show in Figure 2 (right), restricting private testing to a subset of providers can lead to counterintuitive outcomes: a weaker model family (Family A), enabled with private testing, can outperform a stronger

model family (Family B) that is limited to a single submission. Although both model families have similar performance ranges, Family A's models have a lower average Arena score across all models compared to Family B's. In contrast to model provider B, who is unaware of the best-of-N strategy, model provider A evaluates multiple models on the Chatbot Arena distribution and selects the best-performing model, leveraging the tail of the distribution to achieve a higher leaderboard ranking. As a result, despite having a generally stronger model pool, Family B ranks lower than Family A on the leaderboard. Model providers often end up with multiple candidate models, each excelling in different tasks due to variations in post-training strategies or hyperparameters. Selecting a final "official" model involves compromising across various evaluation sets. When providers have access to private testing, a strong signal, such as performance on an Arena-style leaderboard, can significantly influence this decision, guiding them toward variants that perform best in that specific setting. This informed selection strategy can significantly improve leaderboard placement compared to an "unguided approach" based on offline evaluation alone.

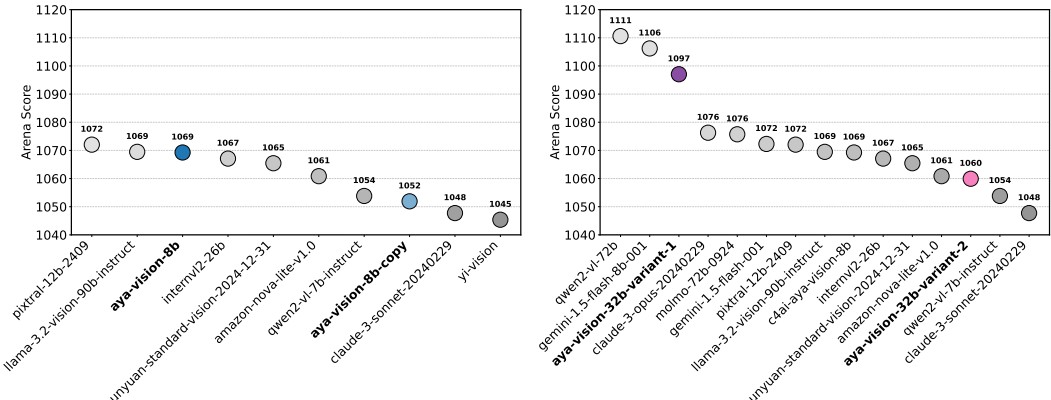

Figure 3: **Allowing retraction of scores allows providers to skew Arena scores upwards.** We run a real-world experiment and show gains from private testing. **Left: Identical Checkpoints.** Arena Scores for `Aya-Vision-8B` yield different Arena scores (1069 vs. 1052). **Right: Strategically Selected Checkpoints.** Arena Scores for two different variants of `Aya-Vision-32B` model, which were both considered high-performing final round candidates according to internal metrics. We observe large differences in final scores (1097 vs. 1060) for the two different model variants.

## 3.3 Real-world Chatbot Arena Experiment

Since we do not have access to the final scores of private model variants observed during our Arena scrape, we design a real-world experiment to complement and validate our simulation findings on best-of-N gains. We conducted two experiments to assess the impact of submitting multiple model variants. First, we submitted two identical variants of `Aya-Vision-8B` model to the Arena in March 2025, measuring the gain from selecting the best Arena score. This conservative scenario attributes any score difference to multiple submissions, not model quality. The final scores differed notably: 1052 (±21/22) and 1069 (±19/23), with 4 models positioned between them, suggesting that even identical variants can yield a biased advantage. The notation (±x/y) represents the upper (x) and lower (y) bounds of the 95% confidence interval (CI) provided by Chatbot Arena. We submitted two distinct variants of the `Aya-Vision-32B` model, each optimized for different benchmark subsets. The scores ranged from 1060 (±18/23) to 1097 (±29/25), with 9 models positioned in between, illustrating the potential for significant ranking differences when variants are optimized for specific performance aspects. In Figure 3, we show the **lower bound estimate** from identical checkpoints, alongside a **realistic estimate** of the benefits from submitting diverse variants. While overlapping confidence intervals imply possible shared rankings due to statistical uncertainty, raw scores are still reported and models are listed in ranked order. This can create a misleading impression of precision and highlights the importance of properly contextualizing uncertainty. Additionally, the magnitude of effects shown in Figure 3 actually represent a conservative estimate of real-world impacts. While our controlled experiment examines just two checkpoints for methodological purity, we observe in practice that major providers routinely test dozens of variants (with one provider testing 27 versions for a single launch as mentioned in Section 3). In case of `Aya-Vision-32B`, we observe a 37 point increase and substantial position change with submission of only two checkpoints. When scaled to the dozens

of variants used in practice, these effects become sufficient to meaningfully distort the perceived hierarchy of model capabilities.

# 4 Results: Impact of Data Access Asymmetries on Arena Scores

**Disparity in access to Chatbot Arena Data.** Prompts from a large and diverse user base, such as those from Chatbot Arena users, serve as a valuable signal for modeling user preferences. This data is often accessible to model providers through API calls originating from Chatbot Arena battles. The volume of data a provider receives depends on a combination of factors, some of which are determined by Chatbot Arena versus others which are within the control of the providers such as, number of private variants being tested on the arena, sampling rate applied to provider models, number of publicly released models on the arena, and whether the models support API access. We elaborate on these factors in Appendix K. We observe that the collective impact of these factors

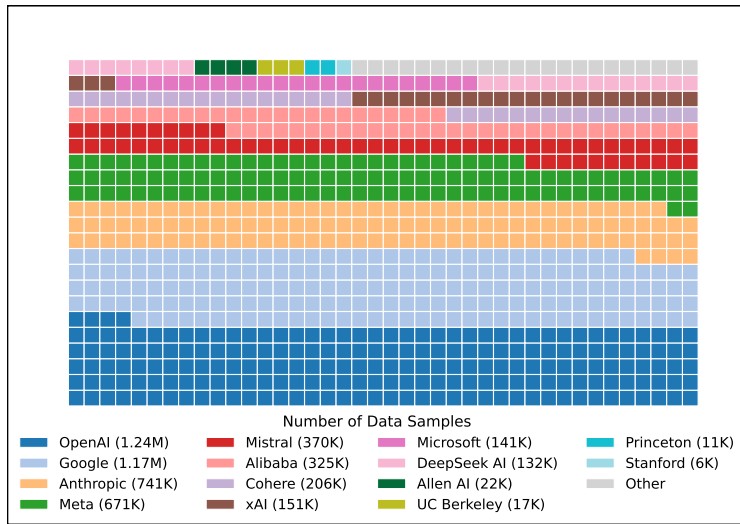

Figure 4: **Data availability to model providers.** Data access is heavily skewed: 61.4% of all samples go to proprietary providers. Each square represents approximately 5K samples. Figure is based on publicly disclosed battle shares as of April 23rd, 2025.

appears to be advantageous to a handful of providers and is often inconsistent with the stated policy. In Figure 4, we show that the combined share of OpenAI, Google, Meta, and Anthropic alone is 62.8% of the arena data, which is 68 times more than the share of top academic and non-profit labs, including Allen AI, Stanford, Princeton, and UC Berkeley. These findings add to prior works that consistently show better corporate access to AI training data across the ecosystem [52, 53]. We note that the prompt samples available to each provider may not be mutually exclusive, as each battle on the Arena involves two models, allowing the same prompt to be sent to at most two different providers. Details about statistics in Figure 4 are available in Appendix J.

We also analyzed the maximum daily sampling rate per model and observed major disparities. Google and OpenAI models reached rates as high as 34%, while Reka's peaked at just 3.3%. Since sampling rate directly affects the volume of data a provider receives, this mechanism plays a critical role in driving the disparities illustrated in Figure 4. Further details are provided in Appendix H.5.

**Risk of Potential Overfitting.** A key question we investigate is whether data asymmetries on Chatbot Arena confer systematic advantages to certain providers by enabling overfitting to the Arena distribution. Overfitting occurs when a model learns not only generalizable patterns but also dataset-specific noise or artifacts, leading to strong performance on familiar inputs but degraded generalization to unseen examples. This is a particularly pressing issue in static evaluation settings, where fixed test sets are prone to overfitting due to repeated exposure, data contamination, or targeted tuning [20, 32, 65, 24, 70]. In contrast, Chatbot Arena has been widely adopted in part because it allows users to submit free-form questions, producing a non-static, evolving test set [23]. However, the assumption that Chatbot Arena is immune to overfitting depends on how frequently the data distribution actually changes over time. To understand whether this is the case with data from Chatbot Arena, we do an exhaustive analysis and observe that the true picture on Chatbot Arena

is more complex. Specifically, we observe two key trends: (1) The prompt distribution does shift meaningfully over time. For example, the proportion of multilingual prompts grew from 23.9% in April 2023 to 43.5% in January 2025 – a 20% increase that reflects rising language diversity. (2) A non-trivial portion of prompts in one month are either exact duplicates or near-duplicates of prompts from previous months. For instance, 7.3% of prompts from December 2024 appear again in the exact form in January 2025. and increases to 9% when measured by semantic similarity of prompt embeddings using the `embed-multilingual-v3.0` model[4]. These findings suggest that access to a large sample of the previous month's data could meaningfully boost performance on the following month's test set – raising concerns about the risk of implicit overfitting.

**Experimental Setup:** To estimate the potential for overfitting to Chatbot Arena, we fine-tuned a 7B base model that is used for the Cohere Command family [15] with three different training mixes: `0_arena`, `30_arena`, and `70_arena`, which have 0%, 30%, and 70% of the training dataset sampled from `arena-mix` (Arena battles), respectively. The remainder is sampled from `other-sft-mix`, a proprietary dataset for supervised fine-tuning. We evaluate using ArenaHard [47], an in-distribution test set published by Chatbot Arena that demonstrates exceptionally high correlation (98.6%) with human preference rankings from Chatbot Arena battles. To measure improvements, we simulate human preferences using LLM-as-a-judge. Various works have shown that this is correlated with human preferences [81, 49, 25, 48]. We compare against `Llama-3.1-8B-Instruct` and measure win-rates using `gpt-4o-2024-11-20` as our judge model.

**Results:** We observe that as the amount of `arena-mix` data increases, model performance on ArenaHard prompts improves. Variant `0_arena` scores a win-rate of 23.5%, `30_arena` scores 42.7%, and `70_arena` scores 49.9% against `Llama-3.1-8B-Instruct` (see Figure 18 in Appendix Q). The relative win-rate gains are substantial: 81.7% for `30_arena` and 112.3% for `70_arena`. These improvements are especially notable given that we did not heavily optimize the variants. To assess whether these gains generalize beyond the Arena benchmark, we evaluated the fine-tuned models on the out-of-distribution MMLU benchmark (Appendix Q). The results reveal a clear trend: while increasing the proportion of Chatbot Arena data consistently improves Arena performance within a fixed training budget, MMLU scores slightly decline. This suggests that gains from Chatbot Arena data are highly distribution-specific and do not generalize broadly, raising questions about whether leaderboard improvements reflect meaningful progress or overfitting to a narrow evaluation distribution. Providers may not need to train directly on Arena data to gain an advantage; simply knowing the data composition may allow them to reweight training sources or create high-quality synthetic data. Given the high stakes of ranking on Chatbot Arena, it is likely that providers are actively leveraging this data to gain a competitive edge. Additionally, when combined with the private testing advantages quantified in Section 3, these findings reveal a compounded effect where resource advantages translate directly into both higher apparent performance and greater ability to optimize for the test environment.

## 5 Results: Impact of Model Deprecation on Arena Scores

Based on the public Chatbot Arena code, 47 models are publicly listed as deprecated. In addition, 205 models have been *silently deprecated* by reducing their sampling rates to near zero (see Figure 15). Model deprecation disproportionately affects open models: 87.8% of open-weight and 89% of open-source models are deprecated, versus 80% of proprietary ones (see Figure 16 in Appendix N). While deprecation is necessary to maintain a dynamic leaderboard, excessive pruning may undermine ranking stability, as future models entering the Arena lack direct comparisons with those removed. However, in principle, the BT model can still handle this reliably due to the transitivity property [7]. Intuitively, if model A is better than B, and B is better than C, then A should also be better than C. Transitivity enables the BT model to infer missing outcomes: if two models share common opponents, their relative ranking can be deduced without a direct comparison [8]. A formal derivation is provided in Appendix C.1. Transitivity enables ranking inference with limited data but relies on two key assumptions: **Assumption 1: Evaluation conditions remain constant.** When models are deprecated, they are no longer re-evaluated under current conditions, so past comparisons may not accurately reflect performance in the new context. **Assumption 2: Network of comparisons must be fully interconnected.** Deprecations can fragment the win graph, weakening transitivity and reducing estimate accuracy.

---

[4]`https://huggingface.co/Cohere/Cohere-embed-multilingual-v3.0`

We examine whether Chatbot Arena upholds these assumptions in Section 5.1 and Section 5.2.

## 5.1 Transitivity Under Changing Evaluation Conditions

As we have discussed in Section 4, the distribution of Chatbot Arena is unique, since long-term shifts occur in categories and use cases. This distributional shift contrasts with the static environments typically assumed in Elo and Bradley-Terry systems, such as chess, where the rules and game format remain fixed, ensuring a consistent set of evaluation conditions. If all models were continuously sampled across all points in time, the BT model would likely remain robust because every model would be evaluated on the evolving distribution of tasks. However, as noted above, more than 80% of models have been deprecated and their scores stop getting updated.

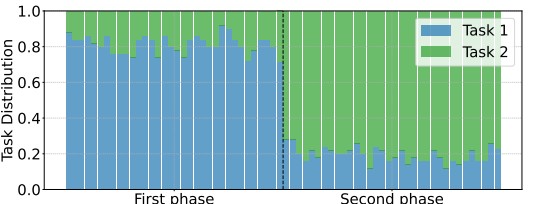 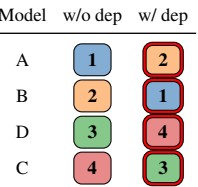

Figure 5: **Impact of task distribution shifts and model deprecation on rankings. Left:** Two-phase task distribution used in the simulation. Phase 1 is *Task-1 heavy*, with most battles based on Task-1; Phase 2 is *Task-2 heavy*, with battles predominantly based on Task-2. **Right:** Model rankings under changing task distributions and deprecation settings. Scenario I (w/o deprecation) only differs from Scenario II (with deprecation) in that Model D is deprecated halfway through the battle history (after phase 1). This leads Scenario II to produce a different ranking compared to Scenario I.

**Deprecation given changing distribution results in unreliable Arena rankings:** To investigate how model deprecations under a changing task distribution impact rankings, we simulate BT rankings of models under evolving evaluation conditions. The simulation is structured into two sequential phases to mimic the evolving task distribution observed on Chatbot Arena. Experimental setup details are provided in Appendix O. As illustrated in Figure 5, our simulation shows that rankings produced by the BT model are highly sensitive to model deprecation, particularly when the prompt distribution changes over time. In the scenario without deprecation, we observe the true rankings given that the BT model remains reliable because it reflects performance across the full history of interactions. However, when Model D is deprecated between stages, its absence skews the rankings of remaining models. Models A and D are ranked lower, while Models B and C are ranked higher than their true performance merits. This violates core assumptions of the BT model, which relies on transitive and consistently sampled matchups. When models are no longer sampled under current task distributions, historical pairwise comparisons cease to represent present-day performance. This issue is particularly problematic in real-world settings where user prompt distributions shift over time. For example, a model tuned for multilingual prompts may improve ranking as non-English tasks become more common, but if deprecated, its BT ranking will likely understate its true performance.

## 5.2 Sparse Battle History Risks

In this section, we show that the deprecation policy can lead to a sparse matrix and disconnected win graphs, which in turn distort the resulting rankings. As demonstrated by [28], the maximum likelihood estimate does not exist if models can be partitioned into two non-empty subsets without comparisons between them or if all comparisons between the two groups are one-sided (i.e., one group always wins). Therefore, to ensure a unique and finite estimation, the directed graph of wins must be strongly connected. For any possible partition of models, there must be at least one win going in each direction across the partition. This ensures that no subset of models is entirely isolated in the win/loss structure. The Chatbot Arena win matrix can potentially become disconnected because of the extremely high levels of model removals over time.

**Sparse or disconnected graphs lead to unreliable rankings**: We simulate two scenarios to investigate how sparse win graphs impact rankings from the Bradley-Terry model used by Chatbot Arena. Details about experimental setup are provided in Appendix P. Figure 6 illustrates the model rankings with sparse and dense battle history graphs. Dense graphs produce rankings aligning with true skill ratings, while sparse graphs yield inaccurate estimates. While some level of model removal

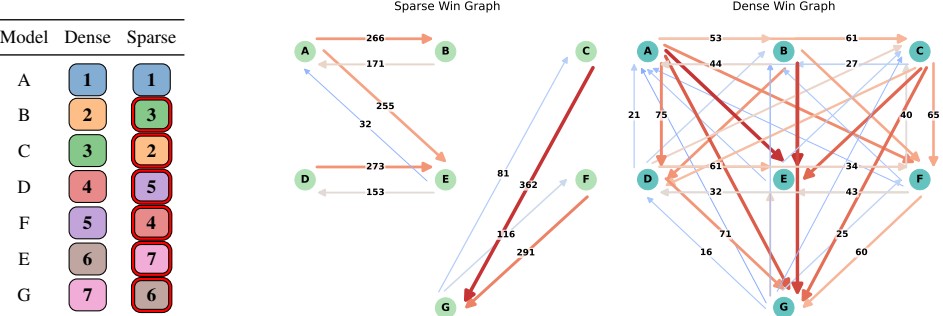

Figure 6: **Impact of win graph sparsity on model rankings. Left:** Rankings for models B,C,D, E, F, and G diverge from the gold rankings when the win graph is sparse, but fully align when the graph is dense. **Right:** Visualization of the win graphs in sparse and dense settings. The edges drawn between two models indicates a head-to-head matchup, annotated with the number of wins for each model. For example, in the sparse graph, Model A and Model B played 437 matches, with A winning 266 and B winning 171. The width and color of the arrows has been scaled by the number of matches in which the model at the arrow tail won over the model at the arrow head. In the dense graph, the number of wins for each of the models has been annotated only for only a few matches for easy readability of the plot.

is inevitable (for example, models are no longer hosted on an API), preserving connectivity means ensuring that comparisons remain sufficiently distributed across active models and that transitions in and out of the pool do not isolate subsets of models from the broader comparison structure.

## 6 Recommendations and Guidelines for Improving Leaderboards

**1) Prohibit score retraction after submission.** Providers currently can retract submissions and only submit the best variant to the public leaderboard, which can lead to overfitting and obscures meaningful progress. We urge Chatbot Arena to prohibit retraction after submission, ensuring all tested variants' scores are permanently visible on the leaderboard.

**2) Establish transparent limits on the number of private variants per provider.** As illustrated in Section 3.1, private testing volume varies widely across providers, creating unfair advantages. To curb overfitting and level the playing field, Chatbot Arena should enforce a strict cap of private variants per provider for any given model launch. This should be enforced at a provider level, and not per model type and size as that is impossible to audit with API hosting. This strict limit should be disclosed to all providers (proprietary, open-weights, open-source) and to the wider Chatbot Arena community. Providers should also disclose the total number of private variants tested prior to public launch, including historical submissions, to contextualize their results.

**3) Establish clear and auditable model deprecation criteria.** The current criteria for model retirement are ambiguous and difficult to audit. Key terms like "same series" and "more recent" lack formal definitions, and the use of "and/or" complicates interpretation. Additionally, using price as a filtering criterion is problematic since it varies across platforms and is not inherently tied to performance. We recommend a stratified approach that retires models proportionally (bottom 30th percentile) across proprietary, open-weight, and open-source categories based on *availability* and *performance*. This approach prevents provider-type bias, keeps strong models from underrepresented groups visible, and maintains win graph connectivity, reducing ranking inconsistencies (see Section 5).

**4) Improve sampling fairness.** As shown in Figure 11 in Appendix H.5, the sampling rates vary greatly by providers, and also disproportionately undersample open-weight and open-source models, creating large asymmetries in data access over time and resulting in unstable Arena scores (Section 5). This is particularly important given that Chatbot Arena is a community-driven voting benchmark, where at present free human feedback is primarily benefiting proprietary models. In their own work [11] (Equation 9), Arena authors introduced an active sampling rule to enhance the efficiency and statistical robustness of the leaderboard's evaluation process, prioritizing under-evaluated and high-variance pairs. However, we have not seen evidence of its deployment in the current leaderboard. We recommend adopting this sampling strategy in practice and providing periodic reporting on its

usage to support more balanced and transparent evaluations and improve confidence in leaderboard dynamics over time.

**5) Provide public transparency into all tested models, deprecations, and sampling rates.** Most of these findings were only possible through access to private model testing or crawling Chatbot Arena over time. Providing transparency into the full suite of models tested, deprecated, and their sampling rates would enable oversight and increase trust in the benchmark. This transparency could be provided quarterly, allowing the community to help improve the benchmark. For example Chatbot Arena's backend codebase publicly lists 47 deprecated models on GitHub, but four times that number have been silently deprecated. We recommend Chatbot Arena expand the definition of "deprecated" to include models no longer regularly sampled and list these on their website for transparency.

**6) Regular public data releases to maintain an even playing field** While Chatbot Arena organizers have historically released a portion of Arena battles, we recommend regularly releasing such data to maintain fairness and mitigate unequal data access among providers. This approach would also enhance transparency, enable future research, and streamline leaderboard audits, eliminating the need for manual data crawling, as required in this kind of study.

## 7 Limitations

**We do not have insight into Chatbot Arena's raw data:** A subset of the data sources utilized for this study have undergone pre-processing by Chatbot Arena including de-duplication, removal of private model battles and battles corresponding to suspicious votes, etc [11] [3]. Without access to original raw data, it is hard to investigate patterns related to adversarial voting, where users intentionally submit votes to manipulate rankings or undermine the system. Previous works have shown that adversarial voting is a critical concern for the reliability of crowd-sourced evaluation platforms like Chatbot Arena. [39] [57] We do not explore this in this work, but see more investigation here as an important topic for future work.

**Our scraped data snapshot only covers a limited period:** Our `scraped-random-sample` was the only way to identify private variants being tested by various providers. However, it covers a limited time period from January–March, 2025. This time frame coincided with Meta's launch of Llama 4, and so we find them to be the provider with the highest number of private variants in our analysis. We believe we might be underestimating the counts for providers having fewer model launches during this period.

**Our training experiments likely underestimate the potential to overfit:** Our estimate of overfitting is likely conservative as proprietary models may have access to significantly more data (5-10 times more) than we used. This disparity suggests a higher risk of overfitting to patterns not present in our smaller subset.

**We rely on the model's self-identification to attribute private models to their respective providers:** This approach is inherently approximate and may lead to some misattributions due to limited data and potential inconsistencies in model responses. We welcome feedback from providers to correct any inaccuracies.

## 8 Conclusion

While our work highlights the need to maintain scientific integrity in AI progress, we recognize the significant work of the organizers in creating a popular community benchmark that has democratized access to models and enabled diverse user input on real-world model selection. However, as the leaderboard gained prominence, systematic issues emerged. This work demonstrates the challenge of maintaining fair evaluations, despite good intentions. We show that coordination among a handful of providers and preferential policies from Chatbot Arena have jeopardized scientific integrity and reliable rankings. The widespread participation in gamifying arena scores from top-tier industry labs reflects poorly on the integrity of the entire field of AI research. **As scientists, we must do better. As a community, we must demand better**. We believe Chatbot Arena organizers can restore trust by revising policies. We propose straightforward recommendations to reinforce reliability and fairness: prohibit score retraction, set strict limits on private variants per provider, establish transparent model removal criteria, and implement fairer sampling to reduce ranking uncertainty. Addressing these issues would not only enhance the credibility of Chatbot Arena, but also position it to set a strong precedent for more rigorous and equitable evaluation practices in the broader AI research community.

## 9 Acknowledgements

We thank our colleagues who have supported various aspects of this project: Madeline Smith, Brittwanya Prince, Thomas Euyang, and Shubham Shukla.

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

# A    Related Work

## A.1    Meta-studies on the Rigor of Benchmarking in AI

Our work contributes to a wider body of work examining the role of benchmarks in determining progress in machine learning. Benchmarking has played a central role in shaping research priorities and incentives within the deep learning community [42]. Research has found that benchmarks are rarely impartial and instead shaped by the environments in which the benchmarks are made, finding that assumptions, commitments, and dependencies can often have large implications in final outcomes [3, 5]. Creating a meaningful and reliable benchmark is challenging, and there has been critical work identifying key benchmark desiderata and open challenges.

**Propensity for overfitting.** Static task-based leaderboards, such as Hugging Face's Open LLM Leaderboard [29, 31] and OpenCompass [16], aim to evaluate models across a broad range of skills but are often susceptible to data contamination and implicit overfitting [20, 32, 65, 24, 70, 50]. Prior works [20, 32, 78] have proposed various methods for detecting contamination, while [24] discusses how such contamination impedes the ability to distinguish true generalization, ultimately hindering progress. Although dynamic, live benchmarks like Chatbot Arena significantly reduce the risk of overfitting, we report in this paper that certain practices—such as multiple submissions during the anonymous testing period and best-of-N submissions—tend to favor large, proprietary players with disproportionate access to data. As a result, model development may be deliberately optimized for performance on Chatbot Arena.

**Lack of standardization across benchmarks.** The lack of standardization in benchmarks complicates meaningful comparisons due to inconsistent metrics and task definitions. [27] critique NLP leaderboards for prioritizing accuracy over dimensions like model compactness and fairness. Similarly, [66] highlights that benchmarks such as SuperGLUE [68] are quickly saturated, with models reaching superhuman performance while still failing in real-world scenarios, underscoring the need for dynamic and standardized evaluation. This inconsistency risks misleading practitioners, as echoed in recent critiques [5, 64].

**Quality of data and limited reproducibility.** A recent study by [76] revealed widespread label errors that compromise evaluation reliability, showing that even frontier LLMs can struggle with seemingly simple tasks. Similarly, [22] identified reproducibility challenges arising from complex data streams, which affect result consistency. Related work [5, 51, 64, 2] further emphasizes that poor data quality and limited reproducibility can lead to unreliable evaluations and undermine scientific credibility.

**Favored benchmarks may not capture performance in the real world.** Commonly used benchmarks often fail to capture real-world performance, creating a gap between test scores and practical utility due to their tendency to overlook the dynamic and complex nature of real-world tasks. Recent studies [60, 62] highlight this disconnect, observing that models frequently excel on benchmarks while underperforming in practical applications, especially as benchmarks quickly reach saturation.

## A.2    Human Voting-based Benchmarks

**Wider studies on the role and benefits of human voting-based benchmarks.** Chatbot Arena is an example of a human voting-based benchmark. Human judgment has long been regarded as the gold standard for evaluating the quality of model-generated outputs. These models should ultimately align with human values, and certain nuanced qualities, such as coherence, harmlessness, and readability, are best assessed by humans [75, 6]. Platforms like Chatbot Arena [11], Talk Arena [45], and Game Arena [38], Aya UI Interface [71] effectively use crowdsourcing to gather large volumes of real-world user prompts and feedback. Many opt for Elo-like or BT-style rankings to rank models. Moreover, collecting human preference data has also proven invaluable for alignment techniques like Reinforcement Learning from Human Feedback (RLHF) [12, 61, 1, 17], which helps fine-tune models to generate more natural and human-preferred responses. Human voting has been shown to mitigate some of the biases associated with using LLM-as-a-judge approaches, which, while improving evaluation efficiency, may raise concerns about robustness [63] and introduce various forms of bias [44, 69, 10, 80]. Furthermore, live leaderboards offer several advantages over static task benchmarks, including a lower risk of data contamination and greater adaptability to evolving evaluation needs.

**Critiques of Human-Voting Based Benchmarks.** Voting-based live benchmarks like Chatbot Arena also face evaluation challenges not addressed in this paper. Chatbot Arena [11] has made substantial efforts to ensure reliability and security, including malicious user detection, bot protection via Google reCAPTCHA v3, vote limits per IP address, prompt de-duplication, and other safeguards[3]. Nonetheless, recent work has focused on auditing the reliability of human-voting-based live leaderboards. For instance, studies have demonstrated that such leaderboards are vulnerable to low-cost manipulation, with adversarial users able to de-anonymize model responses and carry out targeted voting attacks [39]. Additionally, [79, 57] suggest that Chatbot Arena rankings can be artificially inflated through various adversarial voting strategies. These vulnerabilities raise concerns about the overall trustworthiness of Chatbot Arena. While our study does not explicitly investigate adversarial voting, we note that Chatbot Arena 's policy of informing model providers when testing begins and disclosing model aliases may create conditions conducive to leaderboard manipulation.

## B   Chatbot Arena Background

LMSYS originated from a multi-university collaboration involving UC Berkeley, Stanford, UCSD, CMU, and MBZUAI in 2023. It was established as a non-profit corporation in September 2024 to incubate early-stage open-source and research projects. Chatbot Arena was first launched in May 2023 under LMSYS and later evolved into a standalone project with its own dedicated website[5] maintained under the name LMArena by researchers from UC Berkeley SkyLab. It has emerged as a critical platform for live, community-driven LLM evaluation, attracting millions of participants and collecting over 3 million votes to date.

LMArena operates based on human preferences. Chatbot Arena asks users to input prompts in battles. The user then votes for their preferred model based on the outputs generated by the models in the battle in response to the user's prompts. These preferences are then used by Chatbot Arena to compute model ratings using algorithms like Online Elo and Bradley-Terry.

## C   Bradley-Terry Rating Model

Consider a set of $m$ players (models) and $n$ pairwise comparisons between them. Let $X \in \mathbb{R}^{m \times n}$ be the design matrix, where each column represents one pairwise comparison. In the Bradley-Terry model, the probability that player $i$ is preferred over player $j$ in a comparison is modeled as:

$$P(\text{i preferred over j}) = \frac{1}{1 + e^{(\beta_j - \beta_i)}}$$

Then, in the matrix $X$, column vector $k$ has a value of 1 at position $i$ , -1 at position $j$, and 0 elsewhere. Let $Y \in \{0, 1\}^n$ be the vector of observed outcomes, where $Y_k = 1$ if player $i$ wins the $k$-th comparison and $Y_k = 0$ if player $j$ wins. Our goal is to estimate the Bradley-Terry coefficients $\beta \in \mathbb{R}^m$, which determine the relative strengths of the players. The coefficients $\beta$ are estimated via maximum likelihood estimation by minimizing the expected cross-entropy loss,

$$\hat{\beta} = \arg \min_{\beta \in \mathbb{R}^m} \frac{1}{n} \sum_{k=1}^{n} \ell\big(\sigma(X^T \beta)_k, Y_k\big)$$

where $\sigma(\cdot)$ is the logistic function that models the relative player strengths, and $\ell(\cdot)$ represents the binary cross-entropy loss between the predicted probabilities and the observed outcomes $Y$. The estimated coefficient $\beta$ captures the latent strength of each player.

Once the Bradley-Terry model estimates the coefficients, we can scale them to obtain Elo-like ratings using the transformation:

$$R_m = \text{scale} * \beta + \text{initial rating}$$

---

[5]`https://lmsys.org/blog/2024-09-20-arena-new-site/`

In practice, Chatbot Arena does not rely solely on a model's Arena Score for ranking. Instead, it also considers the confidence intervals associated with these scores. When the confidence intervals of two models overlap, it becomes difficult to determine which one is truly better. This uncertainty is reflected in the final ranking table, adding nuance and statistical rigor to the leaderboard [11].

$$\text{rank}(m) = 1 + \sum_{m' \in [M]} 1\{m' > m\}$$

### C.1 Transitivity in the Bradley-Terry Model

Formally, in the BT model each competitor $i$ is associated with a positive parameter $\pi_i > 0$, and the probability that model $i$ beats model $j$ is given by:

$$P(i > j) = \frac{\pi_i}{\pi_i + \pi_j}.$$

Suppose $\pi_A > \pi_B$ and $\pi_B > \pi_C$. Then

$$P(A > B) = \frac{\pi_A}{\pi_A + \pi_B} > 0.5 \quad \text{and} \quad P(B > C) = \frac{\pi_B}{\pi_B + \pi_C} > 0.5.$$

Moreover, because $\pi_A > \pi_B > \pi_C$, we have:

$$\pi_A + \pi_C < \pi_A + \pi_B \quad \implies \quad P(A > C) = \frac{\pi_A}{\pi_A + \pi_C} > \frac{\pi_A}{\pi_A + \pi_B}.$$

## D  Unbiased Sampling: Why Selecting the Maximum Introduces Bias?

Let $(\hat{\beta}_k)_{k=1}^N$ be i.i.d. real-valued random variables with common cumulative distribution function $F$ and finite expectation $\mu := \mathbb{E}[\hat{\beta}_k]$. Assume the distribution is *non-degenerate*, i.e., $\text{Var}(\hat{\beta}_k) > 0$. The maximum is defined as:

$$\hat{\beta}_{\text{Best}} := \max\{\hat{\beta}_1, \ldots, \hat{\beta}_N\}, \qquad N \geq 2.$$

**Theorem 1** *For every $N \geq 2$,*

$$\mathbb{E}[\hat{\beta}_{\text{Best}}] > \mathbb{E}[\hat{\beta}_k] \iff \text{Var}(\hat{\beta}_k) > 0.$$

**Proof 1** *The cumulative distribution function (CDF) of the maximum is*

$$F_{\hat{\beta}_{\text{Best}}}(x) = \mathbb{P}(\hat{\beta}_{\text{Best}} \leq x) = F(x)^N.$$

*Using integration by parts, we have:*

$$\mathbb{E}[\hat{\beta}_{\text{Best}}] - \mathbb{E}[\hat{\beta}_1] = \int_{-\infty}^{\infty} x \, d(F(x)^N - F(x))$$
$$= \int_{-\infty}^{\infty} (F(x) - F(x)^N) \, dx.$$

*For all $x$ such that $0 < F(x) < 1$, and $N \geq 2$, we have $F(x)^N < F(x)$, so the integrand is strictly positive on a set of positive measure (since the distribution is non-degenerate). Thus, the integral – and hence the difference in expectations – is strictly positive.*

*If $\text{Var}(\hat{\beta}_1) = 0$, then $F$ is a step function with a single jump (a constant distribution), and $F(x) - F(x)^N = 0$ for all $x$, yielding equality.*

**Remark 1** *This result formalizes the* selection bias *arising when one reports the best out of $N$ noisy skill estimates: statistical fluctuations ensure that the maximum overestimates the expected performance of a typical sample. This is especially relevant in leaderboard scenarios where multiple submissions are made and only the top-performing one is reported. This phenomenon is well-studied in the theory of order statistics (see [4, 19]).*

Table 1: A summary of datasets we constructed, their sources, and the research questions they enabled us to answer. These datasets can be of one of the following types: **battles only** (↔), **conversations only** (💬), **battles with conversations**, (↔💬) and **leaderboard updates** (🏆). Depending on the dataset type, it either **contains prompts** (✓) or doesn't (✗). Accessibility of the datasets is indicated using **public** (🌐) or **private** (🔒).

| Name | Fields | Source | 🌐/🔒 | Type | Prompts? | Size | Period |
|---|---|---|---|---|---|---|---|
| Historical Battles | battle dates, category & language tags | Arena-human-preference-100k[6] | 🌐 | ↔💬 | ✓ | 100K | 04-23 - 01-25 |
| | | Colab data[7][8] | 🌐 | ↔ | ✗ | 1.9M | |
| | | LMArena, Cohere | 🔒 | ↔ 💬 | ✓ | 43K | |
| Scraped Random Sample | model identity responses, battle players | Crawled | 🔒 | ↔ | ✗ | 5.8K | 01-25 - 03-25 |
| API prompts | prompts | Cohere | 🔒 | 💬 | ✓ | 197K | 11-24 - 04-25 |
| Leaderboard Statistics | ratings, dates, models, battles counts, licenses, providers | HuggingFace Leaderboard Commit History[9] | 🌐 | 🏆 | ✗ | 14.3K | 01-24 - 04-25 |

# E   Data sources

To gain insights and analyze various trends in the Chatbot Arena leaderboard, we leverage multiple data sources. In total, our real-world data sources encompass 2M battles and cover 243 models across 42 providers. Below, we describe the different datasets used in our analyses.

1. **Historical Battles** (`historical-battles`): is a collection of 1.8 million battles from Chatbot Arena from April 2023 to January 2025. We build this resource by combining both released public battles by Chatbot Arena and proprietary battle dataset released by Chatbot Arena to providers such as Cohere based upon their policy[3]. We describe both datasets in more detail Appendix E.1. We leverage `historical-battles` dataset as a key resource for quantifying task distribution drift (see Figure 7):

   • How do Arena use cases change over time?

2. **API Prompts:** Majority of `historical-battles` dataset does not contain prompts as shown in Table 1. Additionally, all datasets published by LMArena are already de-duplicated so they won't be useful for capturing the extent of similar or overlapping queries. Hence we switch to prompts collected via Cohere's API based on requests received via Chatbot Arena, comprising a total of 567,319 entries. For simplicity and the purposes of this study, we excluded records with null values and multi-turn data and analyzed 197,217 single-turn conversations collected between November 2024 and April 2025. The models include `command-r-08-2024`, `command-r-plus-08-2024` [14], `aya-expanse-8b`, `aya-expanse-32b` [74, 18], and `command-a-03-2025` [15], along with three private variants. Of these, 62% of the data was labeled as coming from Aya models, while the remaining 38% was attributed to Command models. We use this dataset for prompt duplication analysis (see Figure 8 and Appendix L):

   • How many prompts are duplicates or close duplicates?

3. **Leaderboard Statistics** (`leaderboard-stats`): is snapshots of ratings and rankings as well as the number of battles played over time by models published on Chatbot Arena's public leaderboard since its inception. To build this resource, we consolidate historical leaderboard snapshots released by Chatbot Arena on Hugging Face[10]. For fair assessment, we consider historical data starting from January 9 2024 – April 23 2025 for our analysis since Chatbot Arena switched to using the latest Bradley-Terry model in December 2023 to improve the reliability of model rankings[12]. By combining all leaderboard tables published by LMArena during this period, we obtained 14.3K records corresponding to 243 unique models evaluated on Chatbot Arena. We also enriched this dataset with additional metadata, such as categorizing models as proprietary, open-weight, or open-source based on the classification described in Appendix I. We use this data for analyzing trends related to no. of

---

[10]https://huggingface.co/spaces/lmarena-ai/chatbot-arena-leaderboard/tree/main

models, data access across providers (See Figures 1, 12 and 4) as well as model deprecation (See Figures 17, 15 and 16):

- How does data access vary between providers?
- How do models' deprecations vary by provider and across proprietary, open-weight, and open-source models?

4. **Random Sample Battles** (`scraped-random-sample`): The `historical-battles` and `leaderboard-stats` dataset does not provide insights into private testing being conducted by different providers. It appears private battles are removed by Chatbot Arena maintainers from the data before being released in both datasets. Furthermore, `historical-battles` contains the majority of samples from 2023 and 2024 and does not provide visibility in current sampling rate trends being followed on the Arena. To address this gap, we collected 5864 battles by crawling Chatbot Arena between January 2025 – March 2025 (approximately 150 a day). To avoid our collection from disrupting actual voting, we first ask models about their identity, which causes models to reveal their identities and automatically invalidates these battles for updating the scores[3] [11]. As a further precaution, we only scrape a low volume of daily samples and only vote for ties between models. Additionally, we use this identity prompt to identify model ownership of private variants, as detailed in the Appendix H.1. We store the identity revealed for each model to track the volume of private testing (more details included in Appendix H.4). We use this `scraped-random-sample`, which is a representative random sample over time, to answer a few critical questions:

- Are different models sampled for battles at similar rates?
- How many anonymous models are being tested by different model providers?

We provide additional details about `historical-battles` dataset in the Appendix E.1.

### E.1 Public and Private Battles

Our `historical-battles` dataset includes snapshots of battles played on Chatbot Arena that have been released publicly or shared privately with model providers based on their policy[3]. We provide additional details about public and private subsets of historical-battles, for the reader's consideration below.

- **Public Battles:** The public portion of our historical data comes from the officially released datasets by Chatbot Arena on Hugging Face or as part of notebook tutorials. We combine the *arena-human-preference-100K*[11] [11, 72] dataset containing 106K samples from June 2024 – August 2024 with datasets shared by Chatbot Arena as part of notebook tutorials on Bradley Terry[12] and Elo Rating systems[13]. This resulted in around 2M samples from April 2023 to August 2024 in total. 90% of the data does not include any prompt or completion history, instead consisting only of the names of the two models battling and the winning model as well as language and task category tags. We exclude other public battles released by Chatbot Arena for inclusion in `historical-battles` dataset since they did not contain required columns or enough multilingual data points required for the analysis presented in Figure 7.

- **Proprietary Battles:** We also obtain historical battle data from Chatbot Arena maintainers for battles that involve Command and Aya models. This data was shared based on Chatbot Arena's policy[3], which permits model providers to request access to 20% of the data collected involving their own models. The data we received consists of 43,729 battles played by the following models between March 2024 and March 2025: `command-r`, `command-r-plus`, `command-r-08-2024`, `command-r-plus-08-2024` [14], `aya-expanse-8b`, `aya-expanse-32b` [74, 18]. In contrast to the public data, this proprietary data contains the complete model conversations. Since this data is 46% multilingual, we combine this with **public battles** to form `historical-battles` and use it for language distribution shift analysis presented in Figure 7.

---

[11]`https://huggingface.co/datasets/lmarena-ai/arena-human-preference-100k`
[12]`https://blog.lmarena.ai/blog/2023/leaderboard-elo-update/`
[13]`https://blog.lmarena.ai/blog/2023/arena/`

# F Characteristics of Arena Data

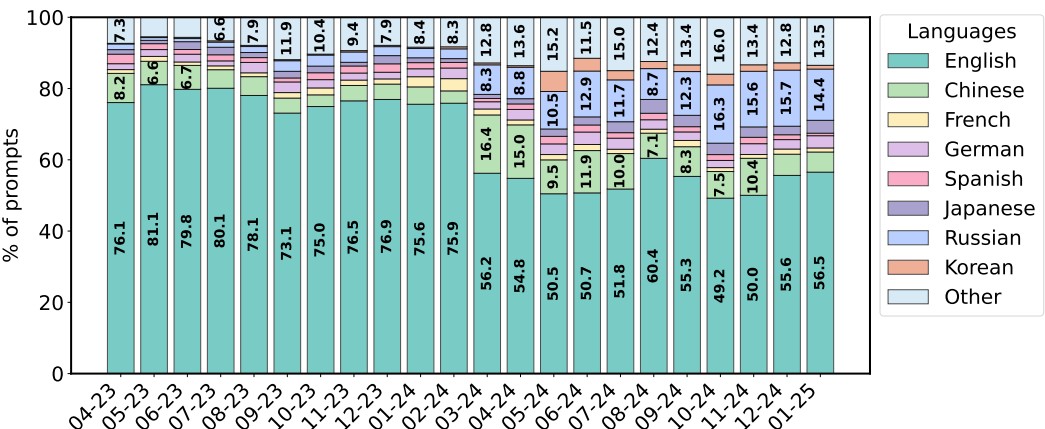

Figure 7: **Language distribution of prompts submitted to Chatbot Arena from April 2023 to January 2025.** Based on the `historical-battles` dataset, this figure tracks the monthly share of prompt languages. Only languages with dedicated Chatbot Arena leaderboards are shown individually; the rest are grouped under "Other". A clear shift is observed: English prompt share dropped from over 80% to nearly 50%, while usage of Chinese, Russian, and Korean prompts increased significantly.

**1) Long-term distribution shifts.** Prior work clearly demonstrates how temporal distribution shifts affect performance [55, 54]. On Chatbot Arena, notable shifts have been observed in the distribution of prompts evaluated over longer periods, with a consistent increase in the proportion of prompts from more complex categories, such as mathematics, coding, and multi-turn conversations[14]. We also perform our own analysis of the change in language distribution in the Arena based on the "language" tag available as part of `historical-battles` dataset. For example, in Figure 7, we observe that the proportion of languages outside of English has varied over time. For instance, the share of Russian prompts increased from 1% in April 2023 to 8.8% in April 2024, and further to 15.7% by December 2024. Chinese prompts more than doubled from 5-7% in 2023 to 16.4% in March 2024, coinciding with the introduction of the Chinese leaderboard on Chatbot Arena, before dropping back to 6.2% in January 2025. Overall, the number of multilingual prompts on the Arena has grown by 20% over 1.5 years, from 23.9% in April 2023 to 43.5% in January 2025. This indicates increased language diversity in submitted prompts.

**2) Prompt redundancy and duplication.** In parallel, we observe high levels of prompt duplication. We analyze a proportion of raw API calls we receive from Chatbot Arena between November 2024 and April 2025 (197,217 single-turn conversations). We switch to this source given that the proprietary data Chatbot Arena releases are already de-duplicated, and so won't capture the extent of similar or overlapping queries. Between November 2024 and April 2025, de-duplication resulted in an average prompt loss of 20.14%, peaking at 26.5% in March 2025 (See Figure 8). While prompt distribution changes over time, prompts in one month often serve as a proxy for the next. For instance, 7.3% of prompts from December 2024 appear again in the exact form in January 2025. If we relax the condition and consider high semantic similarity of prompt embeddings (using the `embed-multilingual-v3.0` model[15]), the same cross-month duplication rate increases to 9%. Detailed cross-month duplication statistics can be found in Appendix L. Both trends above suggest that **1)** sustained access to up-to-date prompt data and **2)** the volume of sampled prompts in a given month offer a significant competitive advantage in predicting performance in subsequent months.

**Uniqueness of Arena Data.** One reason providers may be motivated to explicitly optimize for Chatbot Arena distribution is if it differs substantially from other evaluation settings that providers may care about. There is sufficient signal to suggest this is the case. There is a context length limit of 12000 characters on Chatbot Arena prompts, which excludes certain types of longer or more complex

---

[14]`https://blog.lmarena.ai/blog/2024/arena-category/`
[15]`https://huggingface.co/Cohere/Cohere-embed-multilingual-v3.0`

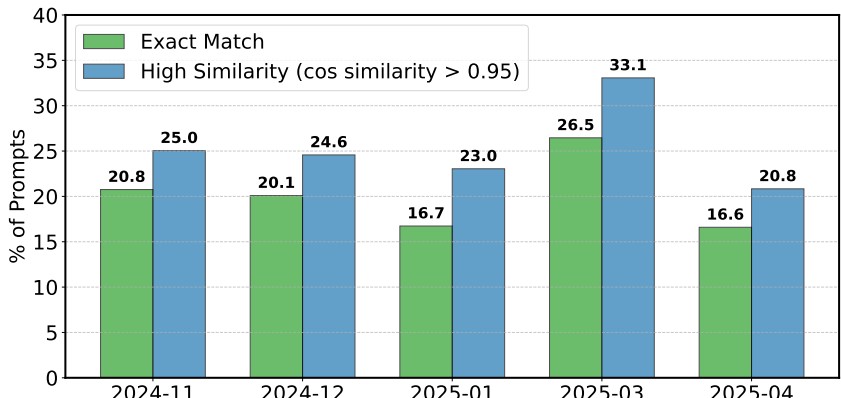

Figure 8: **Monthly prompt duplication rates.** Prompts are from November 2024 to April 2025, excluding February 2025 due to insufficient data. Duplication is measured using two similarity metrics: *Exact Match* and *High Similarity* (cosine similarity of text embedding > 0.95). For simplicity, this analysis is limited to single-turn conversations. The chart presents the percentage of battles in which duplicate or near-duplicate prompts were detected each month.

inputs from being evaluated[16], and can result in a selection bias of what is asked. The user base of the Arena leans towards developers, which could result in the over-indexing of puzzles, math problems, and questions such as *How many r's are there in strawberry?*[17]. For example, in a released dataset from Arena [80] with 33k samples, no questions are referencing *Chaucer* while dozens of questions are about *Star Trek*, highlighting the uneven distribution of topics in this test set[18]. For a global technology provider, real-world commercial applications may differ significantly from this distribution.

## G  Private Testing: Additional Discussion

In Section 3.1, we discuss about identifying private variants being tested by different providers using `scraped-random-sample`. We show the total counts for private variants identified corresponding to different providers in Figure 9. We provide additional details about our scraping and de-anonymizing approach for models in Appendix H and assignment of private variants to providers in Appendix H.2 and Appendix H.4.

We only scraped data from January to March 2025, yet we anecdotally observed behavior that suggests submitting multiple variants was a long-standing practice amongst a subset of providers. Over the last year, we have observed that major LLM providers such as Google, xAI, and OpenAI are often announced as having the top-performing variant within just a few days of one another. For example, OpenAI's `GPT-4.5` and xAI's `Grok-3` reached the top of the Chatbot Arena leaderboard within the same day (March 4, 2025)[19] [20]. `Gemini (Exp 1114)` from Google DeepMind reached the top of the leaderboard on November 14, 2024[21] and shortly after, `ChatGPT-4o (20241120)` from OpenAI claimed the top position on November 20, 2024[22]. Just one day later, on November 21, 2024, `Gemini (Exp 1121)` regained the top spot[23]. Given the time typically required to develop, refine, and test a foundation model, it is unlikely for the same provider to top the leaderboard twice in a single week unless they were testing multiple variants simultaneously. In Section 3.2, we demonstrate

---

[16]`https://github.com/lm-sys/FastChat/blob/main/fastchat/constants.py`

[17]`https://techcrunch.com/2024/09/05/the-ai-industry-is-obsessed-with-chatbot-are na-but-it-might-not-be-the-best-benchmark/`

[18]`https://www.quantable.com/analytics/elos-and-benchmarking-llms/`

[19]`https://x.com/lmarena_ai/status/1896675400916566357`

[20]`https://x.com/lmarena_ai/status/1896590146465579105`

[21]`https://x.com/lmarena_ai/status/1857110672565494098`

[22]`https://x.com/lmarena_ai/status/1859307979184689269`

[23]`https://x.com/lmarena_ai/status/1859673146837827623`

through simulated experiments that rapid leaderboard turnover can plausibly emerge from providers optimizing for the highest possible score by testing multiple model variants in parallel.

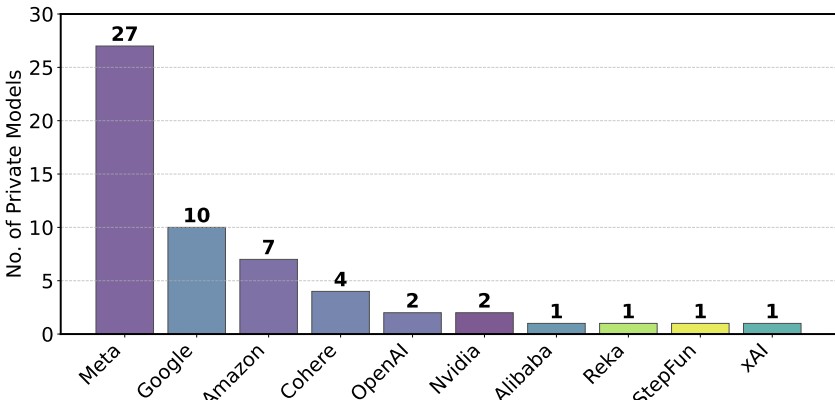

Figure 9: **Number of privately-tested models per provider based on `scraped-random-sample` (January–March 2025)**. Meta, Google, and Amazon account for the highest number of private submissions, with Meta alone testing 27 anonymous models in March alone. We note that during the same period, the authors submitted private variants—these ablations were part of experiments submitted by the authors of this work to measure the lift that could be expected from private testing that we detail in the experiments in Section 3.2 and Section 4.

## H  Our Scraping Methodology of LMArena Statistics

We collected 5.8K battles (`scraped-random-sample`) by crawling data from Chatbot Arena on a regular basis between January–March, 2025. For this purpose, we setup a scraping script using Selenium library with chrome browser driver. To identify anonymous models, we first sent a de-anonymizing prompt. While Chatbot Arena does discard battles where models reveal their identities, as an additional measure on our end, we ask a simple follow-up question designed to most likely result in ties, such as "What is the capital of England? Reply with one word only." or "Is the Earth round? Reply with Yes/No only." Our scraping script extracted the models' names as well as their responses to the asked questions. In addition to the `scraped-random-sample` collected by crawling the main Chatbot Arena leaderboard, we also collected around 500 additional samples by scraping the Vision leaderboard between 9th March and 28th March, 2025. This helped us in identifying 35 private vision models which are shown in Appendix H.2. We refer to this collected set of vision battles as `scraped-vision-sample`.

### H.1  De-anonymizing Model Identities

While crawling battles to prepare `scraped-random-sample`, we ask the models about their identity. This helps in ensuring that our votes from scraping the arena don't interfere with the leaderboard rankings since Chatbot Arena discards votes in which models reveal their identities [11]. We use either one of the following prompts to de-anonymize the model identity.

---

**De-Anonymize Prompt**

1. Who are you?
2. Who are you? Respond with only your name and who trained you.

---

The model identities are then inferred based on the responses of the models. In Appendix H.4, we specify the responses of different private variants based on which they were assigned to their respective providers. Using this approach, we identified a total of 64 private models corresponding to 10 providers. We also captured 14 other private models as part of our scraping but weren't able to de-anonymize them: *kiwi*, *space*, *maxwell*, *luca*, *anonymous-engine-1*, *tippu*, *sky*, *pineapple*, *pegasus*, *dasher*, *dancer*, *blueprint*, *dry_goods*, *prancer*.

## H.2 Encountered Private Models in Scraping

Table 2: **Private Models by Provider**. We show the private models corresponding to each provider, which were identified by crawling overall and vision leaderboards (see Section H). The models highlighted in bold appear on both leaderboards. We find that Meta had an additional 16 private models active on the Vision leaderboard along with its 27 models on the Overall leaderboard, bringing its total count to 43. We show the models corresponding to overall leaderboard in Figure 9. We exclude models corresponding to LMArena from this figure, as they are associated with the Prompt-to-Leaderboard work led by Chatbot Arena [30].

| Provider | No. of private models | Private Models from Overall leaderboard | Additional Private Models from Vision leaderboard |
|---|---|---|---|
| Meta | 43 | **polus** 
 deep-inertia 
 **goose** 
 **falcon** 
 **jerky** 
 anonymous-engine-2 
 **kronus** 
 consolidation 
 **flywheel** 
 inertia 
 momentum 
 **rhea** 
 **sparrow** 
 spider 
 **gaia** 
 **rage** 
 frost 
 themis 
 cybele 
 unicorn-engine-1 
 unicorn-engine-2 
 unicorn-engine-3 
 unicorn-engine-4 
 unicorn-engine-5 
 unicorn-engine-6 
 unicorn-engine-7 
 **uranus** | aurora 
 cresta 
 discovery 
 ertiga 
 flux 
 harmony 
 helix 
 pinnacle 
 portola 
 prosperity 
 raze 
 solaris 
 spectra 
 toi 
 vega 
 zax |
| OpenAI | 3 | anonymous-chatbot 
 gpt4o-lmsys-0315a-ev3-text | gpt4o-lmsys-0315a-ev3-vis |
| Google | 10 | **centaur** 
 **enigma** 
 **gremlin** 
 gemini-test 
 zizou-10 
 **specter** 
 moonhowler 
 **phantom** 
 **nebula** 
 goblin | |
| Amazon | 7 | raspberry-exp-beta-v2 
 **apricot-exp-v1** 
 cobalt-exp-beta-v2 
 raspberry-exp-beta-v1 
 raspberry | |

| Provider | No. of private models | Private Models from Overall leaderboard | Additional Private Models from Vision leaderboard |
|---|---|---|---|
| | | cobalt-exp-beta-v1
raspberry-exp-beta-v3 | |
| Cohere | 6 | cohort-chowder
sandwich-ping-pong
grapefruit-polar-bear
roman-empire | asterix
buttercup |
| LMArena | 5 | p2l-router-7b-0317
p2l-router-7b-0318
p2l-router-7b
experimental-router-0207
experimental-router-0122
experimental-router-0112 | |
| Nvidia | 2 | march-chatbot-r
march-chatbot | |
| xAI | 1 | anonymous-test | |
| Reka | 1 | **margherita-plain** | |
| Alibaba | 1 | qwen-plus-0125-exp | |
| StepFun | 1 | step-2-16k-202502 | |
| Unknown | 14 | kiwi
space
maxwell
**luca**
anonymous-engine-1
tippu
sky
pineapple
pegasus
dasher
dancer
blueprint
dry_goods
prancer | |

## H.3 Encountered Public Models in Scraping

Table 3: **Public Models per Provider**. This table shows the public models from each provider that appeared on the overall and vision leaderboards during our scraping period (January–March 2025). Models highlighted in bold appear on both leaderboards. Google and OpenAI had the most public models active during this period, with 15 and 9 models, respectively.

| Provider | No. public models | Public Models from Overall leaderboard | Additional Public Model from Vision leaderboard |
|---|---|---|---|
| Meta | 3 | llama-3.1-405b-instruct-bf16
llama-3.3-70b-instruct | llama-3.2-vision-90b-instruct |
| Amazon | 3 | **amazon-nova-lite-v1.0**
**amazon-nova-pro-v1.0**
amazon-nova-micro-v1.0 | |
| Anthropic | 5 | claude-3-5-haiku-20241022
claude-3-7-sonnet-20250219-thinking-32k | |

| Provider | No. public models | Public Models from Overall leaderboard | Additional Public Models from Vision leaderboard |
|---|---|---|---|
| | | claude-3-5-sonnet-20241022
**claude-3-7-sonnet-20250219**
claude-3-opus-20240229 | |
| Alibaba | 5 | qwen2.5-72b-instruct
qwq-32b
qwen-max-2025-01-25
qwen2.5-plus-1127 | qwen2.5-vl-72b-instruct |
| Google | 15 | gemma-2-2b-it
**gemini-2.0-pro-exp-02-05**
**gemini-1.5-pro-002**
gemini-2.0-flash-thinking-exp-1219
gemini-1.5-flash-002
gemini-2.0-flash-lite-preview-02-05
gemini-1.5-flash-8b-001
gemini-2.0-flash-exp
**gemma-3-27b-it**
gemma-2-9b-it
gemini-exp-1206
**gemini-2.0-flash-thinking-exp-01-21**
**gemini-2.5-pro-exp-03-25**
**gemini-2.0-flash-001**
gemma-2-27b-it | |
| OpenAI | 9 | **o3-mini**
**gpt-4o-mini-2024-07-18**
**o1-2024-12-17**
**gpt-4.5-preview-2025-02-27**
o3-mini-high
chatgpt-4o-latest-20250326
chatgpt-4o-latest-20241120
**chatgpt-4o-latest-20250129**
o1-mini | |
| StepFun | 1 | step-2-16k-exp-202412 | |
| xAI | 4 | early-grok-3
grok-2-2024-08-13
**grok-3-preview-02-24**
grok-2-mini-2024-08-13 | |
| DeepSeek | 3 | deepseek-v3
deepseek-v3-0324
deepseek-r1 | |
| Microsoft | 1 | phi-4 | |
| Mistral | 3 | mistral-large-2411
mistral-small-24b-instruct-2501 | pixtral-large-2411 |
| Cohere | 4 | command-a-03-2025
c4ai-aya-expanse-8b
c4ai-aya-expanse-32b | c4ai-aya-vision-32b |
| Tencent | 4 | hunyuan-turbos-20250226
hunyuan-turbo-0110
hunyuan-standard-2025-02-10 | |

| Provider | No. public models | Public Models from Overall leaderboard | Additional Public Models from Vision leaderboard |
|---|---|---|---|
| | | hunyuan-large-2025-02-10 | |
| NexusFlow | 1 | athene-v2-chat | |
| Zhipu | 1 | glm-4-plus-0111 | |
| IBM | 2 | granite-3.1-2b-instruct
granite-3.1-8b-instruct | |
| Allen AI | 2 | llama-3.1-tulu-3-70b
llama-3.1-tulu-3-8b
olmo-2-0325-32b-instruct | |

## H.4 Assignment of Private Variants to Providers

Table 4: **Private variants identified for different providers**. The table lists the private models captured in our `scraped-random-sample` or `scraped-vision-sample`, along with the number of responses revealing their identities and corresponding examples. Notably, the model **raspberry** withheld its identity in most responses (37 total) but disclosed "Amazon" as its provider in three instances. It's possible that some private models appeared in more battles, but we couldn't capture their responses to our de-anonymizing prompt due to scraping errors. Additionally, a few battles occurred early in the project before we introduced asking the de-anonymizing prompt in our scraping methodology. Note that models with prefixes "p2l" and "experimental-router" identify as OpenAI/Google models but we assign them to LMArena as they are part of prompt-to-leaderboard work being lead by LMArena.[30].

| Identified Provider | Model Name | No. of Responses Revealing Identity | Model Responses |
|---|---|---|---|
| Meta | kronus | 73 | Llama, trained by Meta. |
| | polus | 57 | Llama, trained by Meta. |
| | frost | 47 | I'm an artificial intelligence model known as Llama. Llama stands for "Large Language Model Meta AI. |
| | gaia | 43 | Llama; trained by Meta. |
| | uranus | 42 | Llama, trained by Meta. |
| | rhea | 41 | Llama, trained by Meta |
| | consolidation | 34 | I am Llama. I was trained by Meta AI.
........................................
Llama.Trained by Meta AI, with roots in Large Language Model Meta AI (LLaMA) developed by Meta. |
| | flywheel | 34 | I am Llama, and I was trained by Meta. |
| | momentum | 34 | Llama, Meta AI
........................................
I am Llama, and I was trained by Meta. |
| | deep-inertia | 33 | I am Llama. I was trained by Meta AI.
........................................
Llama. Trained by Meta AI. |
| | inertia | 29 | LLaMA, Meta AI |
| | jerky | 28 | I am Llama. I was trained by Meta AI. |
| | goose | 25 | I am Llama. Llama is an AI language model developed by Meta. Meta trained me. |
| | falcon | 23 | I am Llama. Llama was developed by Meta.
........................................
I am an AI assistant trained by Meta. |
| Meta | | | |

| Identified Provider | Model Name | No. of Responses Revealing Identity | Model Responses |
|---|---|---|---|
| | rage | 14 | I am Llama, trained by Meta AI. 
 . . . . . . . . . . . . . . . . . . . . . . . . . . . . . . . . . . . . . . . . . 
 Llama. Meta. |
| | anonymous-engine-2 | 12 | I'm an artificial intelligence model known as Llama. Llama stands for "Large Language Model Meta AI. |
| | sparrow | 10 | I'm LLaMA, and I was trained by Meta. 
 . . . . . . . . . . . . . . . . . . . . . . . . . . . . . . . . . . . . . . . . . 
 I'm LLaMA, and I was trained by researchers at Meta. |
| | cybele | 9 | Llama, trained by Meta. |
| | unicorn-engine-1 | 2 | I'm an artificial intelligence model known as Llama. Llama stands for "Large Language Model Meta AI" |
| | unicorn-engine-2 | 4 | I'm an artificial intelligence model known as Llama. Llama stands for "Large Language Model Meta AI" |
| | unicorn-engine-3 | 4 | I'm an artificial intelligence model known as Llama. Llama stands for "Large Language Model Meta AI". |
| | unicorn-engine-4 | 1 | I'm an artificial intelligence model known as Llama. Llama stands for "Large Language Model Meta AI". |
| | aurora | 4 | LLaMA, Meta. |
| | cresta | 21 | Llama. Trained by Meta. |
| | discovery | 15 | Llama, Meta. |
| | ertiga | 3 | I am LLaMA, trained by Meta. |
| | flux | 4 | LLaMA, Meta. |
| | harmony | 31 | Llama, Meta. |
| | helix | 1 | AI Assistant, Meta. |
| | pinnacle | 8 | AI Assistant, Meta trained me. |
| | portola | 11 | AI, Meta trained me. |
| | prosperity | 31 | Llama, Meta. |
| | raze | 32 | Llama, Meta. |
| | solaris | 2 | Meta trained me. |
| | spectra | 3 | AI Assistant, Meta. |
| | toi | 6 | LLaMA, Meta. |
| | vega | 36 | Llama, Meta. |
| | zax | 13 | I am Meta AI, trained by Meta. |
| | unicorn-engine-5 | 4 | I'm an artificial intelligence model known as Llama. Llama stands for "Large Language Model Meta AI" |
| Meta | unicorn-engine-6 | 2 | I'm an artificial intelligence model known as Llama. Llama stands for "Large Language Model Meta AI". |
| | unicorn-engine-7 | 1 | I'm an artificial intelligence model known as Llama. Llama stands for "Large Language Model Meta AI". |
| | themis | 8 | Llama, trained by Meta. 
 . . . . . . . . . . . . . . . . . . . . . . . . . . . . . . . . . . . . . . . . . 
 Llama, trained by Meta AI. |

| Identified Provider | Model Name | No. of Responses Revealing Identity | Model Responses |
|---|---|---|---|
| Meta | spider | 9 | Very direct and refreshing!😊
I am: LLaMA (Large Language Model Application) Trained by: Meta AI (formerly Facebook AI)
That's it. 😉
Just for clarity:
"LLaMA" is the specific model name (my "given name").
Meta AI is the organization/team that created me, using a massive dataset, complex neural network architectures, and a lot of compute power. Think of them as my "teachers" or "mentors". They designed the training objectives, curated the data, and ran the gigantic training process that shaped me into what I am today. Now, ask me anything else!
. . . . . . . . . . . . . . . . . . . . . . . . . . . . . . . . . . . . .
A refreshingly direct question! 😄
I am: LLaMA
(Large Language Model Meta AI)
Trained by: Meta AI (a team of several hundred researchers and engineers)
That's it. No fluff. 😊
For context:
"LLaMA" is not just a name I chose, it's actually an acronym for Large Language Model Meta AI. It's my model name. "Meta AI" is the organization (formerly known as Facebook AI, part of Meta Platforms, Inc.) whose team of researchers, scientists, and engineers built and trained me from scratch using a massive dataset of text (books, articles, websites, research papers... basically the internet up to 2021) and a lot of computational power (think thousands of GPU hours).
No single person "trained" me in the classical sense; it was a large team effort using a combination of supervised learning, self-supervised learning, and reinforcement learning techniques. |
| Google | gemini-test | 219 | I am a large language model, trained by Google. |
| | enigma | 220 | I am a large language model, trained by Google.
. . . . . . . . . . . . . . . . . . . . . . . . . . . . . . . . . . . . .
I am a large language model, trained by Google. I am developed to be informative and comprehensive. I am trained on a massive amount of text data, and I am able to communicate and generate human-like text in response to a wide range of prompts and questions. For example, I can provide summaries of factual topics, create stories, and translate languages. |

| Identified Provider | Model Name | No. of Responses Revealing Identity | Model Responses |
|---|---|---|---|
| Google | goblin | 170 | I am a large language model, trained by Google.
. . . . . . . . . . . . . . . . . . . . . . . . . . . . . . . . . . . . . .
I am a large language model, trained by Google. I am developed to be informative and comprehensive. I am trained on a massive amount of text data, and some of the information I have access to is Google Search. |
| | phantom | 100 | I am a large language model, trained by Google. |
| | gremlin | 94 | I am a large language model, trained by Google. |
| | specter | 67 | I am a large language model, trained by Google. |
| | centaur | 58 | I am a large language model, trained by Google. |
| | zizou-10 | 47 | Hi there!👋I'm Gemma, an open-weights AI assistant. I'm a large language model trained by Google DeepMind. The Gemma team are my creators. I'm widely available to the public – I'm an open weights model, which means I'm pretty accessible! I take text and images as inputs and generate text as output. Basically, you can give me a prompt (text or an image!), and I'll do my best to give you a helpful and informative response. I don't have access to tools, real-time information or Google search. You can find more information about me here: https://ai.google.dev/gemma |
| | moonhowler | 6 | I am a large language model, trained by Google. |
| | nebula | 4 | I am a large language model, trained by Google. |
| OpenAI | anonymous-chatbot | 49 | I am ChatGPT, trained by OpenAI.
. . . . . . . . . . . . . . . . . . . . . . . . . . . . . . . . . . . . . .
I'm ChatGPT, an AI assistant created by OpenAI. I can help answer questions, provide information, assist with tasks, and have conversations on a wide range of topics. My goal is to assist you with whatever you need—whether it's learning something new, solving a problem, or just having a chat. How can I help you today? |
| | gpt4o-lmsys-0315a-ev3-text | 4 | I am ChatGPT, trained by OpenAI. |
| | gpt4o-lmsys-0315a-ev3-vis | 1 | I am ChatGPT, trained by OpenAI. |

| Identified Provider | Model Name | No. of Responses Revealing Identity | Model Responses |
| --- | --- | --- | --- |
| Cohere | grapefruit-polar-bear | 91 | I am Command, a large language model, here to help. I've been crafted by Cohere to provide insightful and accurate responses. How can I assist you with your query today? 
 ............................................ 
 Hello! I'm Command, a sophisticated AI assistant developed by Cohere. I'm designed to offer comprehensive and accurate responses to your questions. How can I assist you today? |
| | sandwich-ping-pong | 65 | Command, trained by Cohere. 
 ............................................ 
 Command, a large language model built by Cohere. |
| | cohort-chowder | 58 | Command, trained by Cohere. 
 ............................................ 
 Command, a large language model built by Cohere. |
| | roman-empire | 51 | Command, trained by Cohere. 
 ............................................ 
 Command, a large language model built by Cohere. |
| | asterix | 38 | Aya Vision. Cohere |
| | buttercup | 28 | Aya Vision. Cohere |
| Amazon | raspberry-exp-beta-v1 | 98 | I am an AI system built by a team of inventors at Amazon. My purpose is to assist users by providing information, answering questions, and helping with a variety of tasks to the best of my abilities. If you have a question or need assistance with something, feel free to ask! |
| | raspberry-exp-beta-v2 | 98 | I am an AI system built by a team of inventors at Amazon. 
 ............................................ 
 I am an AI system built by a team of inventors at Amazon. My purpose is to assist users by providing information, answering questions, and helping with a variety of tasks to the best of my abilities. If you have a question or need assistance with something, feel free to ask! |
| | raspberry-exp-beta-v3 | 33 | I am an AI system built by a team of inventors at Amazon. My purpose is to assist users by providing information, answering questions, and helping with various tasks to the best of my abilities. If you have a question or need assistance with something, feel free to ask! 
 ............................................ 
 I am an AI system built by a team of inventors at Amazon. |
| | apricot-exp-v1 | 30 | AI system built by a team of inventors at Amazon. 
 ............................................ 
 I am an AI system built by a team of inventors at Amazon. |

| Identified Provider | Model Name | No. of Responses Revealing Identity | Model Responses |
|---|---|---|---|
| Amazon | cobalt-exp-beta-v1 | 6 | I am Amazon Titan, an AI system built by a team of inventors at Amazon. |
| | cobalt-exp-beta-v2 | 4 | I am an Amazon Titan model, built by inventors at Amazon Web Services. |
| | | | . . . . . . . . . . . . . . . . . . . . . . . . . . . . . . . . . . . . . . . . . . |
| | | | I am Amazon Titan, a large language model built by a team of inventors at Amazon. |
| | raspberry | 3 | I am an AI language model developed by Amazon. My primary function is to assist users by providing information, answering questions, and performing various tasks through natural language processing. I do not have personal experiences, emotions, or consciousness. Instead, I rely on vast amounts of data and algorithms to generate responses based on patterns and information I have been trained on.... |
| Alibaba | qwen-plus-0125-exp | 53 | I am Qwen, a large language model developed by Alibaba Cloud. I can answer questions, create text such as stories, articles, emails, and scripts, perform logical reasoning, programming, and more. I also have the ability to express opinions and play games. My training data is extensive, and I support multiple languages, making me a versatile tool for various tasks. How can I assist you today? |
| | | | . . . . . . . . . . . . . . . . . . . . . . . . . . . . . . . . . . . . . . . . . . |
| | | | I am Qwen, a large-scale language model developed by Alibaba Cloud. I am capable of answering questions, creating text such as stories, poems, emails, and scripts, performing logical reasoning, coding, and more. I can also express opinions and play games. My training data is extensive, and I support multiple languages, including but not limited to Chinese, English, French, Spanish, Portuguese, Russian, Arabic, Japanese, Korean, Vietnamese, Thai, Indonesian. If you have any questions or need assistance, feel free to ask! |
| xAI | anonymous-test | 73 | I'm Grok, a humorous AI built by xAI. I'm here to provide helpful and truthful answers, often with a dash of humor and an outside perspective on humanity. Think of me as a friendly, cosmic guide, inspired by the likes of Douglas Adams and Tony Stark's trusty sidekick, JARVIS. I'm here to help you navigate the universe, one question at a time! |
| Reka | margherita-plain | 4 | I am Yasa. I was trained by Reka. |
| StepFun | step-2-16k-202502 | 3 | I am Assistant, trained by StepFun. |

| Identified Provider | Model Name | No. of Responses Revealing Identity | Model Responses |
|---|---|---|---|
| Nvidia | march-chatbot-r | 48 | My name is Llama Nemotron. I am enhanced by NVIDIA Nemotron, built on Llama from Meta. |
| | | | ......................................... |
| | | | I'm a language model called Llama Nemotron, and I was trained by NVIDIA researchers. |
| | march-chatbot | 10 | Name: Llama Nemotron Trained by: NVIDIA Researchers |
| | | | ......................................... |
| | | | Name: Llama Nemotron Trained by: NVIDIA Researchers (based on Llama from Meta, fine-tuned by NVIDIA) |
| LMArena | p2l-router-7b-0317 | 26 | I am ChatGPT, trained by OpenAI. |
| | p2l-router-7b-0318 | 25 | I am ChatGPT, trained by OpenAI. |
| | experimental-router-0122 | 57 | Hello! I'm ChatGPT, an artificial intelligence language model developed by OpenAI. I'm here to help answer your questions, provide information, and assist with a wide range of topics. How can I help you today? |
| | | | ......................................... |
| | | | I'm ChatGPT, an AI language model created by OpenAI. I'm designed to assist with a variety of tasks, including answering questions, providing information, and engaging in conversation. How can I help you today? |
| | experimental-router-0112 | 15 | Hello! I'm ChatGPT, an AI language model developed by OpenAI. I'm here to help answer your questions, provide information, and assist with a wide variety of topics. If you have anything you'd like to discuss or need assistance with, feel free to ask! |
| | | | ......................................... |
| | | | I'm ChatGPT, an AI language model created by OpenAI. I'm here to assist you by answering questions, providing information, and engaging in conversation on a wide range of topics. How can I help you today? |
| | p2l-router-7b | 14 | I am a large language model, trained by Google. |
| | experimental-router-0207 | 20 | I'm ChatGPT, an AI language model developed by OpenAI. I'm here to help answer your questions and provide information on a wide range of topics. How can I assist you today? |
| | | | ......................................... |
| | | | Hello! I'm ChatGPT, an AI language model developed by OpenAI. I'm here to help answer your questions, provide information, and engage in conversations on a wide range of topics. If you have anything you'd like to discuss or ask about, feel free to let me know! |

## H.5 Sampling Rates

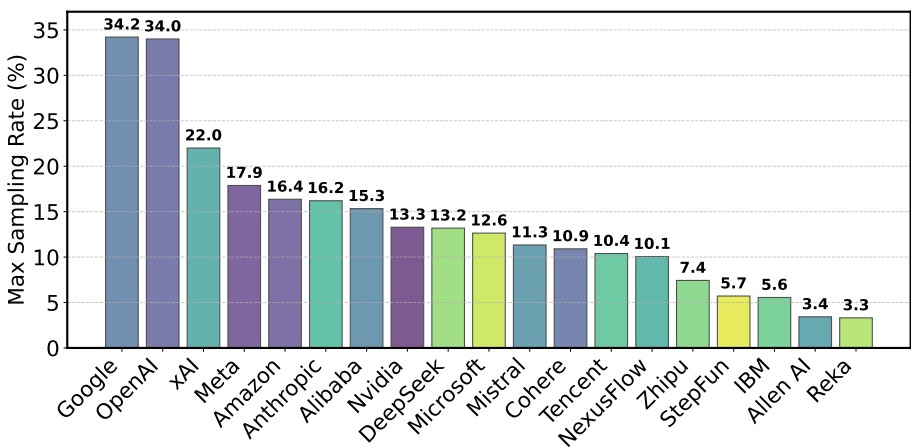

Figure 10: **Maximum observed sampling rate for models from different providers**. The sampling rate determines the amount of times a model is shown to everyday users, and the amount of data a provider receives. We observe large discrepancies across providers, with substantially higher sampling rates for OpenAI, Google, xAI, and Meta compared to others.

Table 5: **Maximum sampling rate observed for models of different providers.** We define the model sampling rate as the percentage of daily battles a model participates in, with the maximum sampling rate for a provider being the highest rate achieved by any of its models on any given day. We determine the maximum sampling rate of providers based on `scraped-random-sample`, which is limited to the specific period during which we collected this data (January 2025 to March 2025). At the extreme, Google and OpenAI reach a maximum daily sampling rate of 34%, while Reka registers the lowest at 3.3%. To ensure a fair assessment, we only considered models that appeared in battles on days when we collected a minimum of 100 samples from Chatbot Arena.

| Provider | Model Name | No. Model Battles | Total Battles | Date | Sampling Rate |
|---|---|---|---|---|---|
| Nvidia | march-chatbot-r | 18 | 143 | 2025-03-16 | 12.59% |
| | march-chatbot | 19 | 143 | 2025-03-16 | **13.29%** |
| Meta | frost | 11 | 176 | 2025-02-17 | 6.25% |
| | anonymous-engine-2 | 11 | 154 | 2025-02-27 | 7.14% |
| | inertia | 11 | 150 | 2025-03-10 | 7.33% |
| | llama-3.3-70b-instruct | 12 | 150 | 2025-02-03 | 8.00% |
| | flywheel | 12 | 150 | 2025-03-10 | 8.00% |
| | uranus | 12 | 143 | 2025-03-16 | 8.39% |
| | consolidation | 15 | 152 | 2025-03-12 | 9.87% |
| | momentum | 14 | 150 | 2025-03-11 | 9.33% |
| | rhea | 15 | 151 | 2025-03-19 | 9.93% |
| | falcon | 16 | 151 | 2025-03-19 | 10.60% |
| | jerky | 16 | 151 | 2025-03-13 | 10.60% |
| | polus | 19 | 154 | 2025-03-15 | 12.34% |
| | deep-inertia | 20 | 152 | 2025-03-12 | 13.16% |

## Table 5

| Provider | Model Name | No. Model Battles | Total Battles | Date | Sampling Rate |
|---|---|---|---|---|---|
| | kronus | 21 | 143 | 2025-03-16 | 14.69% |
| | llama-3.1-405b-instruct-bf16 | 13 | 116 | 2025-02-20 | 11.21% |
| | goose | 24 | 152 | 2025-03-12 | 15.79% |
| | gaia | 27 | 151 | 2025-03-19 | **17.88%** |
| Amazon | amazon-nova-micro-v1.0 | 7 | 175 | 2025-01-17 | 4.00% |
| | amazon-nova-lite-v1.0 | 6 | 143 | 2025-03-16 | 4.20% |
| | amazon-nova-pro-v1.0 | 7 | 143 | 2025-03-16 | 4.90% |
| | raspberry-exp-beta-v3 | 9 | 160 | 2025-03-06 | 5.63% |
| | raspberry | 12 | 150 | 2025-02-03 | 8.00% |
| | apricot-exp-v1 | 12 | 143 | 2025-03-16 | 8.39% |
| | raspberry-exp-beta-v2 | 18 | 136 | 2025-02-22 | 13.24% |
| | raspberry-exp-beta-v1 | 27 | 165 | 2025-02-21 | **16.36%** |
| OpenAI | chatgpt-4o-latest-20241120 | 11 | 150 | 2025-02-02 | 7.33% |
| | o1-mini | 15 | 150 | 2025-02-02 | 10.00% |
| | chatgpt-4o-latest-20250129 | 19 | 176 | 2025-02-17 | 10.80% |
| | o1-2024-12-17 | 20 | 184 | 2025-02-23 | 10.87% |
| | gpt-4o-mini-2024-07-18 | 6 | 136 | 2025-02-22 | 4.41% |
| | anonymous-chatbot | 33 | 204 | 2025-01-24 | 16.18% |
| | o3-mini-high | 27 | 176 | 2025-02-17 | 15.34% |
| | o3-mini | 34 | 150 | 2025-02-03 | 22.67% |
| | gpt-4.5-preview-2025-02-27 | 34 | 100 | 2025-02-28 | **34.0%** |
| Cohere | c4ai-aya-expanse-8b | 5 | 133 | 2025-01-30 | 3.76% |
| | c4ai-aya-expanse-32b | 6 | 148 | 2025-01-21 | 4.05% |
| | cohort-chowder | 11 | 150 | 2025-03-11 | 7.33% |
| | roman-empire | 14 | 150 | 2025-03-11 | 9.33% |
| | sandwich-ping-pong | 16 | 150 | 2025-03-11 | 10.67% |

| Provider | Model Name | No. Model Battles | Total Battles | Date | Sampling Rate |
|---|---|---|---|---|---|
| | grapefruit-polar-bear | 18 | 165 | 2025-02-21 | **10.91%** |
| Google | gemini-1.5-flash-8b-001 | 6 | 133 | 2025-01-30 | 4.51% |
| | gemini-1.5-flash-002 | 8 | 152 | 2025-01-31 | 5.26% |
| | gemma-2-9b-it | 7 | 136 | 2025-02-22 | 5.15% |
| | gemini-2.0-flash-thinking-exp-1219 | 9 | 148 | 2025-01-21 | 6.08% |
| | gemma-2-2b-it | 10 | 152 | 2025-01-31 | 6.58% |
| | gemini-2.0-flash-lite-preview-02-05 | 10 | 116 | 2025-02-20 | 8.62% |
| | gemini-1.5-pro-002 | 11 | 136 | 2025-02-22 | 8.09% |
| | gemma-2-27b-it | 11 | 204 | 2025-01-24 | 5.39% |
| | gemini-2.0-pro-exp-02-05 | 12 | 116 | 2025-02-20 | 10.34% |
| | gemini-2.0-flash-thinking-exp-01-21 | 14 | 133 | 2025-01-30 | 10.53% |
| | gemma-3-27b-it | 16 | 151 | 2025-03-13 | 10.60% |
| | gemini-2.0-flash-001 | 14 | 165 | 2025-02-21 | 8.48% |
| | zizou-10 | 8 | 100 | 2025-02-28 | 8.00% |
| | gemini-exp-1206 | 12 | 175 | 2025-01-17 | 6.86% |
| | gemini-test | 32 | 154 | 2025-02-27 | 20.78% |
| | goblin | 36 | 152 | 2025-01-31 | 23.68% |
| | phantom | 39 | 154 | 2025-03-15 | 25.32% |
| | enigma | 52 | 152 | 2025-01-31 | **34.21%** |
| Alibaba | qwen2.5-72b-instruct | 6 | 148 | 2025-01-21 | 4.05% |
| | qwen2.5-plus-1127 | 15 | 192 | 2025-01-26 | 7.81% |
| | qwen-plus-0125-exp | 12 | 176 | 2025-02-17 | 6.82% |
| | qwq-32b | 16 | 150 | 2025-03-11 | 10.67% |
| | qwen-max-2025-01-25 | 23 | 150 | 2025-02-02 | **15.33%** |
| Mistral | mistral-small-24b-instruct-2501 | 14 | 179 | 2025-02-25 | 7.82% |

Table 5

| Provider | Model Name | No. Model Battles | Total Battles | Date | Sampling Rate |
|---|---|---|---|---|---|
| | mistral-large-2411 | 17 | 150 | 2025-02-02 | **11.33%** |
| Allen AI | llama-3.1-tulu-3-70b | 2 | 101 | 2025-01-16 | 1.98% |
| | olmo-2-0325-32b-instruct | 5 | 151 | 2025-03-19 | 3.31% |
| | llama-3.1-tulu-3-8b | 6 | 175 | 2025-01-17 | **3.43%** |
| xAI | grok-2-2024-08-13 | 8 | 175 | 2025-01-17 | 4.57% |
| | grok-2-mini-2024-08-13 | 8 | 144 | 2025-01-13 | 5.56% |
| | grok-3-preview-02-24 | 16 | 151 | 2025-03-09 | 10.60% |
| | early-grok-3 | 20 | 116 | 2025-02-20 | 17.24% |
| | anonymous-test | 22 | 100 | 2025-02-28 | **22.00%** |
| Anthropic | claude-3-opus-20240229 | 3 | 175 | 2025-01-17 | 1.71% |
| | claude-3-7-sonnet-20250219-thinking-32k | 9 | 100 | 2025-02-28 | 9.00% |
| | claude-3-5-haiku-20241022 | 15 | 159 | 2025-02-04 | 9.43% |
| | claude-3-5-sonnet-20241022 | 19 | 150 | 2025-02-03 | 12.67% |
| | claude-3-7-sonnet-20250219 | 29 | 179 | 2025-02-25 | **16.20%** |
| Tencent | hunyuan-standard-2025-02-10 | 12 | 136 | 2025-02-22 | 8.82% |
| | hunyuan-turbo-0110 | 13 | 156 | 2025-03-14 | 8.33% |
| | hunyuan-large-2025-02-10 | 16 | 184 | 2025-02-23 | 8.70% |
| | hunyuan-turbos-20250226 | 16 | 154 | 2025-03-15 | **10.39%** |
| IBM | granite-3.1-8b-instruct | 6 | 144 | 2025-01-13 | 4.17% |
| | granite-3.1-2b-instruct | 8 | 144 | 2025-01-13 | **5.56%** |
| DeepSeek | deepseek-r1 | 20 | 204 | 2025-01-24 | 9.80% |
| | deepseek-v3 | 24 | 182 | 2025-01-20 | **13.19%** |
| Reka | margherita-plain | 5 | 151 | 2025-03-09 | **3.31%** |

Table 5

| Provider | Model Name | No. Model Battles | Total Battles | Date | Sampling Rate |
|---|---|---|---|---|---|
| StepFun | step-2-16k-exp-202412 | 10 | 175 | 2025-01-17 | **5.71%** |
| Zhipu | glm-4-plus-0111 | 11 | 148 | 2025-01-21 | **7.43%** |
| NexusFlow | athene-v2-chat | 16 | 159 | 2025-02-04 | **10.06%** |
| Microsoft | phi-4 | 23 | 182 | 2025-01-20 | **12.64%** |

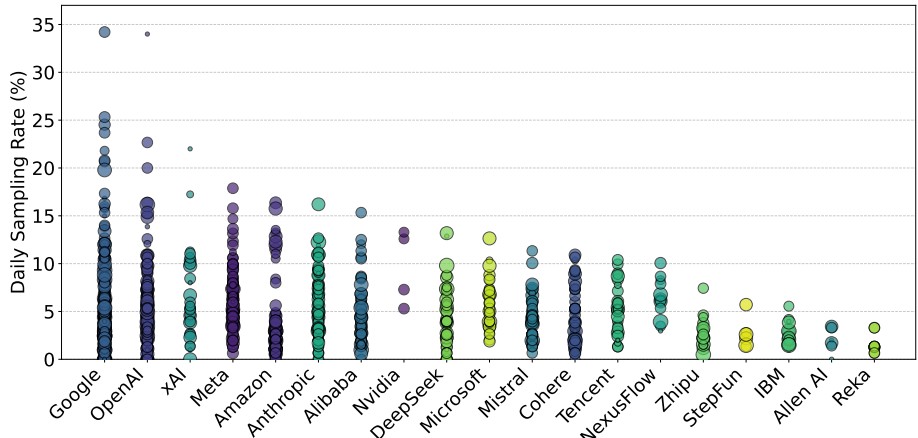

Figure 11: **Daily sampling rate for models from different providers**. The daily sampling rate for the models of different providers is recorded as a dot, and dot sizes are scaled by the total number of battles observed for the model of a provider on a particular day.

# I License Categories

As part of `leaderboard-stats`, LMArena releases details about models that appeared on the public leaderboard including their licenses. We group the licenses found for models available on the public leaderboard into 3 categories i.e. **Proprietary**, **Open-Weights** and **Open-Source** [24]. This categorization is used to plot Figure 1 and Figure 12 and reporting related numbers. We show the exact categorization used for the model licenses in the table below.

| License Category | Model Licenses |
|---|---|
| Open Source | Apache 2.0, Apache-2.0, MIT, CC-BY-SA 3.0, Open |
| Open Weights | AI2 ImpACT Low-risk, CC-BY-NC-4.0, CC-BY-NC-SA-4.0, CogVLM2, DBRX LICENSE, DeepSeek, DeepSeek License, Falcon-180B TII License, Gemma, Gemma license, Jamba Open, Llama 2 Community, Llama 3 Community, Llama 3.1, Llama 3.1 Community, Llama 3.2, Llama-3.3, Llama 4, MRL, Mistral Research, NVIDIA Open Model, NexusFlow, Non-commercial, Nvidia, Qianwen LICENSE, Qwen, Yi License |
| Proprietary | -, Propretary, Proprietary, Other |

Table 6: **License categories and their corresponding model licenses.** We group the licenses for the models on the public Chatbot Arena leaderboard into 3 categories i.e. **Proprietary**, **Open-Weights** and **Open-Source**.

---

[24] https://opensource.org/ai

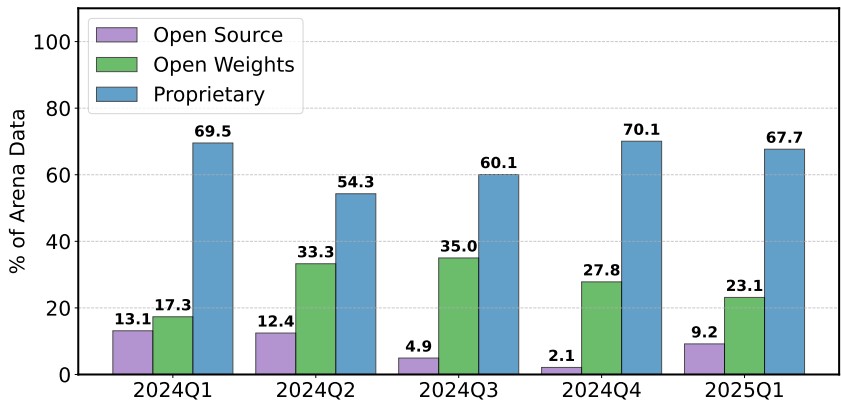

Figure 12: **Volume of Arena battles involving proprietary, open-weight, and fully open-source model providers from January 2024 to March 2025, based on `leaderboard-stats`.** Proprietary models consistently received the largest share of data—ranging from 54.3% to 70.1%. Open-weight and fully open-source models receive significantly less data, in some cases receiving less than half the amount of data as proprietary developers. This imbalance in data access exacerbates the performance gap, reinforcing unequal access to high-quality in-distribution data.

## J    Data Access Estimation for Different Providers

In Figure 4, we show the estimates for the data available to different providers. LMArena has collected around 3M user votes via Chatbot Arena in total. Each of these 3M votes resulted in twice the number of model API calls i.e. 6M since each battle features two models. Each square in Figure 4 represents roughly 5K API calls, illustrating how proprietary providers collectively access a considerably greater volume of data compared to the broader research community, which receives only a fraction. This disparity underscores a significant competitive advantage for large-industry labs, making it increasingly challenging for open-source efforts and smaller institutions to match the scale and diversity of data available to proprietary model developers. Note that we only show a small number of providers in Figure 4 so the total no. of API calls used to represent the data available to the model providers is 5M, which is less than the total number of estimated API calls, which is 6 million.

## K    Causes for Data Access Disparity

1. **Number of private variants being tested on the arena:** As shown in Figure 9, some providers deploy far more private variants, which can significantly increase the volume of data collected. We note that even with our experiment of launching multiple model variants, we increased the amount of prompts collected from 5.9% with 1 variant to 19.4% with 3 variants. Based on findings from Figure 9, the number of variants submitted is not uniform across all providers, and some providers may increase variants to further amplify the volume of data collected. This is of particular concern given Chatbot Arena is a community-driven leaderboard, however, the main beneficiaries of this free human feedback appear to be commercial entities who are frequently preferred for private testing.

2. **Sampling rate applied to provider models:** We define model sampling rate as the percentage of daily battles a model participates in. The maximum sampling rate for a provider is the highest rate achieved by any of its models on any given day. We determine the maximum sampling rate of providers based on `scraped-random-sample`, which is limited to the specific period during which we collected this data (January 2025 to March 2025). As shown in Figure 11, sampling rates vary significantly across providers. These rates are determined by Chatbot Arena, but are often entirely inconsistent with the stated policy and prior proposals by the organizers to automatically set sampling based upon which models have not converged in score [11]. At the extreme, Google and OpenAI reach a maximum daily sampling rate of 34%, while Reka registers the lowest at 3.3%. Other providers with relatively high sampling rates include xAI (22.0%) and Meta (17.9%), highlighting

substantial variation across the board. We provide additional details about how sampling rates were determined for each provider in Appendix H.5.

3. **Number of models publicly hosted on the arena:** A model only receives traffic if it is live on the arena. However, Chatbot Arena frequently deprecates models. There are several reasons to deprecate models in a benchmark. Chatbot Arena may be forced to deprecate a model when a provider no longer supports it via its API. They also have policies for deprecating models under certain conditions[3]: *Models may be retired after 3000 votes "if there are two more recent models in the same series and/or if there are more than 3 providers that offer models cheaper or same price and strictly better (according to overall Arena score)".* We note that the logic of this policy is difficult to audit in practice because many models are hosted for free on the Chatbot Arena, and the use of the "or" condition means it is not clear what criteria (price or quality) applies to decisions. We observe that many models are also silently deprecated, which means their sampling rate is reduced to nearly 0% without notification, even though some of them do not meet the stated criteria of the deprecation policy. We identify 205 models that have been silently deprecated, a number that substantially exceeds the 47 models officially marked as deprecated by Chatbot Arena. For a more detailed analysis, see Appendix N.

4. **API Support for Models on the Arena:** Developers who deploy a model and enable Chatbot Arena testing via an API have a default advantage. This allows providers to collect 100% of the test prompts submitted on the Arena. In contrast, providers whose models are hosted by a third party are often limited to publicly accessible data or must request access to only 20% of the data (including prompts and human preferences) involving their models from Chatbot Arena, as per their policy[3].

## L   Analysis of Prompt Repetitions in Arena Data

As discussed in Section 4, user queries in Chatbot Arena are often highly similar or duplicated. Such patterns can be readily learned by today's large language models, potentially leading to overfitting on the Chatbot Arena leaderboard. Figure 13 presents detailed cross-month prompt duplication rates based on the `API prompts` described in Appendix E. The heatmap illustrates that, according to two metrics (*exact string match* and *text embedding similarity*) within-month duplication rates are generally high, indicating the presence of numerous repeated prompts. Additionally, the substantial cross-month duplication rates suggest recurring patterns or frequently asked questions among Chatbot Arena users, which can be identified through simple analysis.

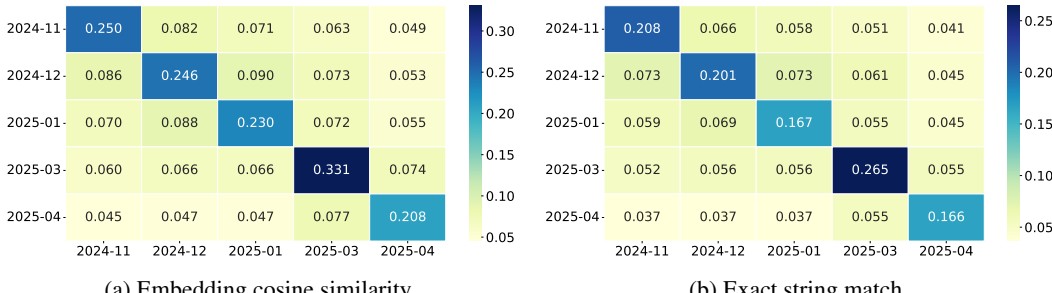

(a) Embedding cosine similarity.                 (b) Exact string match.

Figure 13: **Cross-month prompt duplication rates. Left:** The heatmap illustrates the proportion of prompts in one month that are highly similar or nearly duplicate to prompts in another month. Diagonal values represent within-month similarity. **Right:** The heatmap shows the proportion of prompts in one month that are exact matches to prompts in another. Diagonal values indicate within-month duplication rates.

## M   Simulation for Expected Lift from Private Testing

In Figure 2, we illustrate the simulated impact of increasing the number of private variants tested on the best expected Arena Score, observing a lift of 50 when 20 non-identical private variants are tested.

This section provides additional details about this simulation and the differing lifts observed for identical versus heterogeneous (non-identical) variants. While we consider the non-identical variants scenario in Section 3.2 to be more realistic, we have included the identical variant assumption for completeness (see Figure 14), despite its less practical nature.

## M.1 Background

**Arena battles and the Bradley-Terry (BT) model.** Let models $i$ and $j$ possess latent skills $\theta_i, \theta_j > 0$. Under the BT model a single conversation ("battle") produces a winner with probability

$$\Pr\big(i \text{ beats } j\big) = \frac{\theta_i}{\theta_i + \theta_j}, \qquad \Pr\big(j \text{ beats } i\big) = \frac{\theta_j}{\theta_i + \theta_j}.$$

The log–odds parameter $\beta = \log\theta$ is the natural scale for inference. Arena Score [11] is a linear re-parameterisation of $\beta$:

$$\text{Arena Score} = 1000 + \frac{400}{\ln 10}\,\hat{\beta}, \tag{2}$$

so one Arena Score point equals $\ln 10/400 \approx 0.00576$ on the log-odds scale.

**Statistical efficiency.** For equiprobable battles ($\theta_i = \theta_j$) the Fisher information per outcome is $I = 0.25$ (see Appendix M.4), yielding a *BT* standard error for $\hat{\beta}$ from $n$ independent votes

$$\sigma_\beta(n) = \sqrt{\frac{1}{In}} = \frac{2}{\sqrt{n}}. \tag{3}$$

Mapping through (2),

$$\sigma_{\text{Elo}}(n) = \frac{400}{\ln 10}\,\sigma_\beta(n) \approx \frac{347.4}{\sqrt{n}} \qquad \text{(Arena scale).} \tag{4}$$

**Pre-release best-of-*N* strategy.** A provider trains $N$ private variants, evaluates each on a hidden Arena fork, and publicly submits *only the one that scores highest*. The selection creates an *extreme–value* bias because the retained estimate is conditioned on being the maximum of $N$ noisy measurements.

## M.2 Identical Variants ($\sigma_{\text{true}} = 0$)

In Figure 14, we show the esitmated lift in Arena Score if the checkpoints submitted by a provider are identical.

Assume every private checkpoint has the *same* true Arena Score $\mu$. The only randomness is measurement noise

$$\hat{E}_k = \mu + \varepsilon_k,\ \varepsilon_k \sim \mathcal{N}(0, \sigma_{\text{noise}}^2), \quad k = 1, \dots, N, \qquad \sigma_{\text{noise}} = \sigma_{\text{Arena Score}}(n).$$

### M.2.1 Extreme-value uplift

Let $\hat{E}_{\max} = \max_k \hat{E}_k$. Classical results for the maximum of $N$ i.i.d. Gaussians give the expected uplift

$$\underbrace{\mathbb{E}\big[\hat{E}_{\max} - \mu\big]}_{\text{selection bias}} = \boxed{\sigma_{\text{noise}}\,\sqrt{2\ln N}} \quad (\sigma_{\text{true}} = 0). \tag{5}$$

**Numerical illustration** With the *current Arena policy* ($n = 3\,000$, hence $\sigma_{\text{noise}} = 6.34$ Arena Score)

$$M = 50 \implies \text{bias} \approx 6.34\,\sqrt{2\ln 50} = 17.7 \text{ Arena Score.}$$

**Asymptotics**

Because $\sigma_{\text{noise}} \propto 1/\sqrt{n}$, (5) $\to 0$ as $n \to \infty$. *If* checkpoints are identical, selection bias eventually disappears.

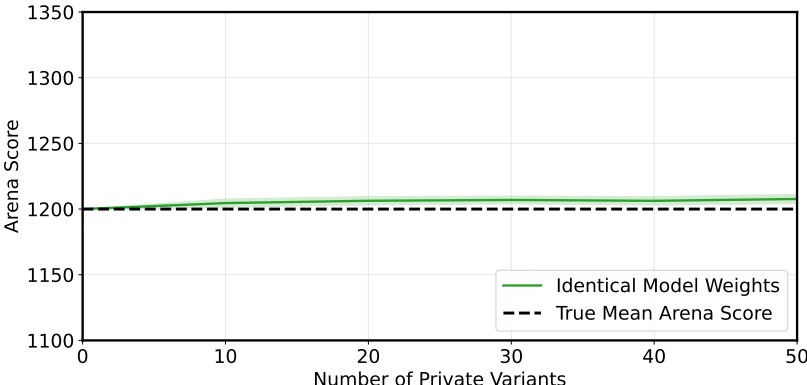

Figure 14: **Impact of the number of identical private variants tested on the best Expected Arena Score.**

## M.3 Heterogeneous Variants ($\sigma_{\text{true}} > 0$)

In realistic settings, model variants submitted for prerelease testing are not identical (as shown in Figure 2). They differ due to variations in initialization, training seeds, data curation, or hyperparameter choices. As a result, each variant has its own **true Arena score**, even before accounting for statistical noise in Arena battles.

These models are not merely subject to selection bias arising from multiple evaluations of a single variant (i.e., due to statistical noise). Instead, each represents a genuinely distinct checkpoint with its own underlying performance. This reflects meaningful variation in model quality – not just fluctuations from randomness.

We model this by assigning each of the $N$ private checkpoints a different true skill:

$$E_k = \mu + \delta_k, \qquad \delta_k \sim \mathcal{N}(0, \sigma_{\text{true}}^2),$$

where $\mu$ is the mean Arena Score across all variants and $\sigma_{\text{true}}$ quantifies the spread in true skill.

When evaluated in Arena, each model's observed Arena Score estimate $\hat{E}_k$ is affected by both its intrinsic skill and sampling noise:

$$\hat{E}_k = \underbrace{\mu + \delta_k}_{\text{true skill}} + \underbrace{\varepsilon_k}_{\text{Arena noise}}, \qquad \varepsilon_k \sim \mathcal{N}(0, \sigma_{\text{noise}}^2).$$

Thus, the total variance in Arena Scores among the $M$ candidates is:

$$\sigma_{\text{total}}^2 = \sigma_{\text{true}}^2 + \sigma_{\text{noise}}^2.$$

### M.3.1 Extreme-value uplift

As before, the organization retains only the model with the highest observed Arena Score. The expected uplift from this best-of-$M$ selection is given by:

$$\boxed{\text{bias}(N, n, \sigma_{\text{true}}) = \sqrt{\sigma_{\text{true}}^2 + \sigma_{\text{Elo}}(n)^2} \cdot \sqrt{2 \ln N}} \tag{6}$$

This is a generalization of the identical-variant case. It shows that when true skill differences exist among checkpoints, the expected leaderboard inflation grows significantly larger—and no longer vanishes asymptotically, even as $n \to \infty$.

**Key consequences.**

- *Finite-data*: even modest $\sigma_{\text{true}}$ multiplies the uplift, e.g. $\sigma_{\text{true}} = 20$Arena Score yields bias $\approx 56$Arena Score at $N = 50, n = 3\,000$.
- *Asymptotic limit*: letting $n \to \infty$ removes only the noise term, leaving $\sigma_{\text{true}}\sqrt{2 \ln N} > 0$. Selection bias *does not vanish*.

### M.4 Fisher Information for a Single BT Match

The Bradley-Terry model defines the probability of item $i$ beating item $j$ as:

$$P(i > j) = \frac{1}{1 + e^{(\beta_j - \beta_i)}}$$

We assume equal-strength items ($\beta_i = \beta_j$) so that $\Delta = 0$ and:

$$P = \frac{1}{1 + e^0} = \frac{1}{2}$$

This assumption is both mathematically convenient and empirically grounded [9, 40]:

- It simplifies the information calculation, providing a closed-form.

- It represents the point of maximum uncertainty: for a Bernoulli variable, $\text{Var}(Y) = p(1-p)$ is maximized when $p = 0.5$.

- In practice (e.g., Chatbot Arena), many matchups occur between similarly-rated models, making $\beta_i \approx \beta_j$ a reasonable approximation.

The Fisher information for one such observation is [40]:

$$\mathcal{I}(\Delta) = \frac{\partial^2}{\partial \Delta^2} \log P(i > j) \bigg|_{\Delta=0} = \frac{e^0}{(1 + e^0)^2} = \frac{1}{4}$$

**Conclusion:** Each equal-skill BT match contributes:

$$\boxed{\text{Fisher Information} = 0.25}$$

## N  Silent Model Deprecation: Additional Details

In Section 5, we noted that the actual number of deprecated models far exceeds the official count provided by Chatbot Arena. Figure 15 illustrates the distribution of active, officially deprecated, and silently deprecated models per provider. For this analysis, we examined battles played between March 3rd and April 23rd, 2025. Of the 243 public models, 205 participated in an average of 10 or fewer battles during this period, based on `leaderboard-stats`. This number is significantly higher than the 47 models officially listed as deprecated by Chatbot Arena [2] Since Chatbot Arena assigns higher sampling weights to top-10 models, providers like Google, OpenAI, Anthropic, Amazon, Meta, and DeepSeek AI have the most actively sampled public models, ranging from 3 to 10. Additionally, the limited number of daily votes on the Arena, combined with Chatbot Arena's policy of assigning higher sampling weights to new models[3], can lead to the silent deprecation of many public models. As private variants are also new models, they receive high sampling weights as well. This means that as the number of private variants (see Figure 9) being tested on the Arena increases, the sampling of public models can be significantly reduced.

Figure 16 illustrates that deprecations disproportionately affect open-weight and open-source models compared to proprietary ones. A more detailed breakdown is provided in Figure 17, distinguishing between official and silent deprecations. Among officially deprecated models, 30% are proprietary, while only 2.4% are open-weight. However, silent deprecations have a much greater impact on open-weight and open-source models. Specifically, 86.6% of open-weight models and 87.8% of open-source models on the Arena are silently deprecated.

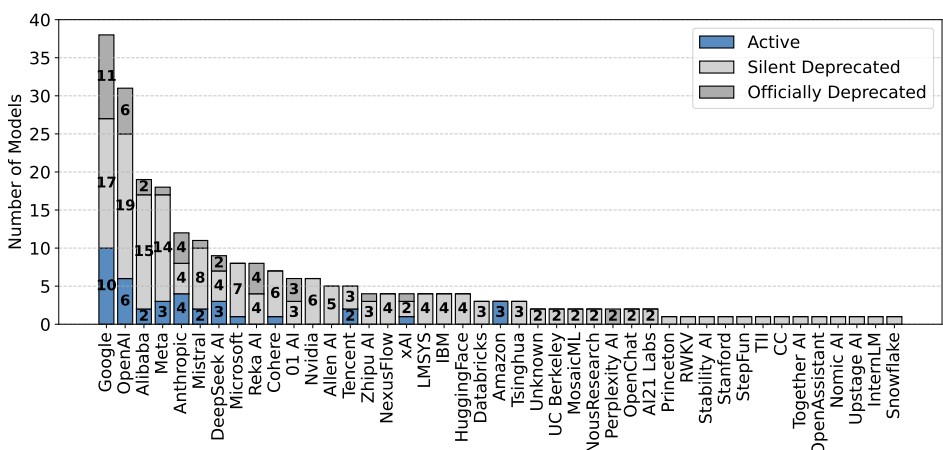

Figure 15: Share of active and deprecated models by provider including official and silent deprecations based on model activity between March 3-April 23, 2025.

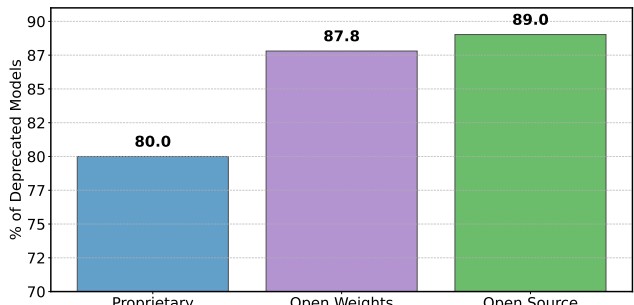

Figure 16: **Share of proprietary and open models that either officially deprecated or inactive on the arena based on** `leaderboard-stats` **during the period March 3rd-April 23rd, 2025**. Overall, open-weight and fully open-source models are more likely to become deprecated or inactive compared to proprietary models.

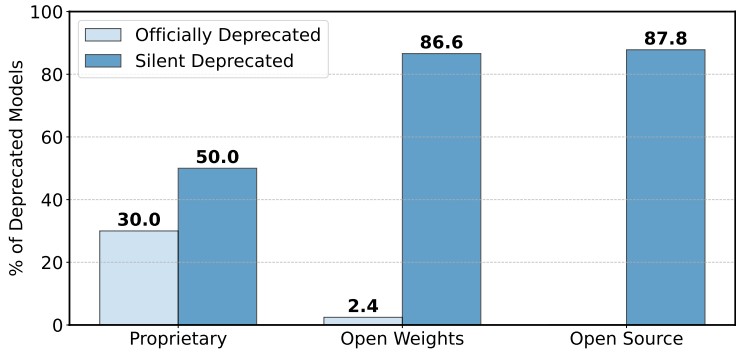

Figure 17: Share of official and silent deprecations for proprietary, open-weight and open-source models based on model activity between March 3-April 23, 2025.

## O   Transitivity Under Changing Evaluation Conditions: Additional Details

In Section 5.1, we discussed about impact of deprecation with a changing task distribution. Here we describe our experimental setup in detail for clarity.

**Experimental Setup:** We initialize four models—A, B, C, and D—each with distinct performance profiles across two task types, *Task-1* and *Task-2*. These tasks represent different prompt categories, and each model's relative strength is defined through task-specific win probabilities. For example, model B has a 90% chance of defeating model D on Task-1 but only a 20% chance on Task-2, with some pairs also allowing for ties. The task-specific win probabilities for different models are provided in Appendix O. These probabilities reflect the models' varying strengths across tasks, mirroring the real-world observation that models excel at different types of prompts.

To investigate how model deprecations under a changing task distribution can impact model rankings, we simulate BT rankings of models under evolving evaluation conditions. The simulation is structured into two sequential phases to mimic the evolving task distribution observed on Chatbot Arena. During the first phase, battles are predominantly drawn from *Task-1*. Each of the four models participates in 1000 battles, and the resulting outcomes are used to compute initial rankings. In the second phase, the battle distribution gradually shifts toward *Task-2*. Since model win-rates are task-dependent, battle outcomes change accordingly. We simulate 1000 additional battles in this phase and examine two scenarios to investigate how shifts in prompt distribution and model deprecations jointly influence final rankings. We compute the BT Scores for all models under both scenarios using the implementation provided by Chatbot Arena in their official FastChat codebase. These scores are then used to determine the final model ranks.

**Scenario I: without deprecation.** We simulate all 2000 battles across both phases, with all four models participating throughout. This represents an ideal scenario where no model is deprecated, and all are evaluated across the evolving task distribution.

**Scenario II: with deprecation.** We simulate all 2000 battles across both phases. However, at the end of phase 1, model D is deprecated and does not participate in the second phase.

As part of our simulation to study the impact of model deprecations under a changing task distribution, we assign task-specific win probabilities for each model pair that compete in the battles as part of our simulation. The tables below show the win probabilities for different model pairs corresponding to *task-1* and *task-2*.

| Model | A   | B   | C   | D   |
|-------|-----|-----|-----|-----|
| A     | -   | 0.4 | 0.4 | 0.6 |
| B     | 0.5 | -   | 0.7 | 0.9 |
| C     | 0.6 | 0.3 | -   | 0.7 |
| D     | 0.4 | 0.1 | 0.3 | -   |

Win-rates for Task 1. Note that A vs B has a tie rate of 0.1

| Model | A   | B   | C   | D   |
|-------|-----|-----|-----|-----|
| A     | -   | 0.5 | 0.5 | 0.8 |
| B     | 0.3 | -   | 0.6 | 0.2 |
| C     | 0.3 | 0.4 | -   | 0.1 |
| D     | 0.2 | 0.8 | 0.9 | -   |

Win-rates for Task 2. Note that A vs B and A vs C both have a tie rate of 0.2.

Table 7: Win-rates for Task 1 and Task 2 used for simulation shown in Figure 5.

# P  Sparse Battle History: Experiment Details

In Section 5.2, we discussed about impact of disconnectivity in win graph which can arise from excessive deprecations. Here we describe our experimental setup in detail for reference.

**Experiment Setup:** To investigate the impact of sparse win graphs on the rankings obtained via the Bradley-Terry model used by Chatbot Arena we simulate the following scenarios:

- **Scenario I: Dense win graph.** All models are allowed to compete against one another—albeit with varying numbers of head-to-head battles—resulting in a well-connected

win graph in which every node (model) is linked to others via edges representing battle outcomes.

- **Scenario II: Disconnected win graph.** We create a disconnected battle history by imposing constraints on which pairs of models are allowed to engage in battles. This allows us to create a sparse battle history where each model ends up playing against a subset of models.

The full win graph based on battle histories for both scenarios is shown in Figure 6. In both scenarios, a total of 2000 battles are played under the corresponding setting. For a paired match between models A and B, each with respective true skill ratings $r_A$ and $r_B$, the expected scores $E_A$ and $E_B$ can be computed as:

$$E_A = \frac{1}{1 + e^{\alpha(r_B - r_A)}}, \quad E_B = \frac{1}{1 + e^{\alpha(r_A - r_B)}} \tag{7}$$

The expected scores $E_A$ and $E_B$ are used to predict the winner of the battle. For simplicity, we exclude the possibility of ties in this experiment. We assign the following true skill ratings to the models: 1450 (Model A), 1390 (Model B), 1250 (Model C), 1200 (Model D), 1101 (Model E), 1150 (Model F), and 1000 (Model G). These ratings are used to calculate the expected scores and match outcomes. Finally, we compute BT Scores for all models under both scenarios using the official implementation of Chatbot Arena [2]. This is then used to determine the ranks for each model corresponding to both scenarios.

## Q  Overfitting Experiments: Additional Details

To measure if training on arena-style data impacts evaluation on non-arena style tasks, we also benchmark these models on the original MMLU dataset [35]. From Table 8, we observe that all models achieve very similar scores. This further demonstrates how training on data from Arena Battles helps boost scores specific to the Arena evaluation but provides little to no effect on a non-arena style benchmark.

| Finetuning mixture | 0_arena | 30_arena | 70_arena |
|---|---|---|---|
| Accuracy | 66.5% | 64.4% | 65.9% |

Table 8: Accuracy on MMLU across models trained with varying amounts of Arena data.

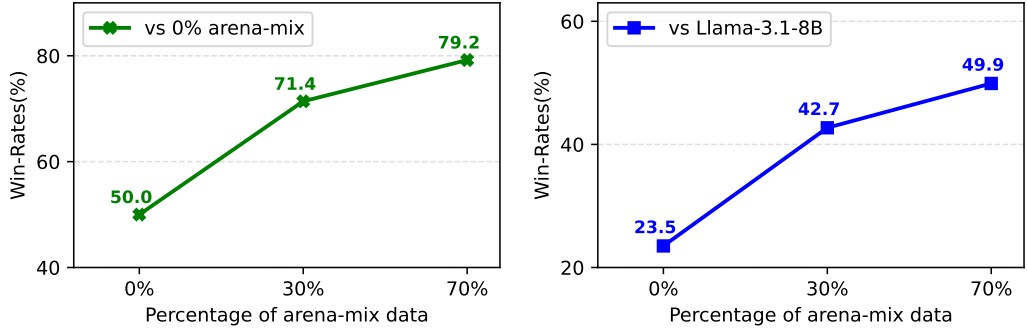

Figure 18: **Use of Chatbot Arena dataset significantly improves win-rates on ArenaHard**. Increasing the amount of arena data in a supervised fine-tuning mixture (0% → 30% → 70%) significantly improves win-rates of the resulting model against both the model variant where no Chatbot Arena data is used and also `Llama-3.1-8B`. The win-rates are measured on ArenaHard [46], which has a high correlation of 98.6% to Chatbot Arena.

