# OpenReview forum: "The Leaderboard Illusion"
_NeurIPS.cc/2025/Datasets_and_Benchmarks_Track — NeurIPS 2025 Datasets and Benchmarks Track poster_

### Official Review · Reviewer_6BEd · 2025-06-15

**Rating:** 5
**Confidence:** 4

**Summary:**

This paper describes an extensive analysis with a set of experiments carried out to identify potential flaws in the commonly-used ChatBot Arena, and follows this up with a set of recommendations to improve the reliability, transparency, accuracy, and fairness of its leaderboard.

**Additional Feedback:**

I would like to point to the following two references, which may be relevant to also discuss. They are not about ChatBot Arena specifically, but more generally about things like overfitting in/to AI benchmarks:

- Roelofs, R., Shankar, V., Recht, B., Fridovich-Keil, S., Hardt, M., Miller, J., and Schmidt, L. A meta-analysis of overfitting in machine learning. In Wallach, H., Larochelle, H., Beygelzimer, A., d'Alch´e-Buc, F., Fox, E., and Garnett, R. (eds.), *Advances in Neural Information
Processing Systems*, volume 32. Curran Associates, Inc., 2019.
- https://arxiv.org/abs/2505.00612

(no, I am not an author or otherwise affiliated with either of them :) )

**Dataset Code Accessibility:**

Partly

**Dataset Code Comments:**

I think code is not available yet, but authors promise that it will be made available later in the Checklist.

**Ethical Considerations:**

No, there are no or only very minor ethics concerns

**Final Justification:**

I have raised my score from Borderline Accept to Accept. This is based on my trust and hope that the authors will be able to clarify and more crisply and cleanly discuss points as per the discussions with reviewers.

**Limitations Weaknesses:**

1. One of the main points in the paper is about how "top providers" with proprietary models (basically: the biggest of the big tech companies) receive more data from ChatBot Arena, enabling them to essentially better "overfit" to the benchmark. Line 56 describes this being a disadvantage for open-source / open-weight models, but really, I suppose it is rather a... disadvantage for entities that are not one of the huge big tech companies? Technically, those companies could also release open-source / open-weight models, and benefit from that same overfitting. Either way, I would like to raise two discussions on this:

    - (a) **is this actually a problem?** I do find the idea of *anyone* at all overfitting to ChatBot Arena specifically, regardless of who does it or how they do it, problematic. But I don't read this as being your main argument (at least, not in these parts of the paper). The current argument in the paper here seems to be mostly one about fairness: it's unfair that top providers (read: big tech companies) get to overfit, whilst smaller entities don't. I'm very much a GPU-poor person myself, and like to occasionally complain about how unfair the world is, but feel like I have to play devil's advocate here. Why should we care about fairness here? It's just one other item in a long list of privileges (they have more GPUs, they have more data outside of ChatBot Arena-specific data, they have more world-class engineers, they have artists helping them make prettier images for their papers, ...). Ultimately, if they can progress science in ways that I can't, because they have resources that I don't... so be it. Too bad for me, good for them. Of course, whether or not they are *actually* progressing science by overfitting to a benchmark is questionable, but again, that does not seem to be the core argument here?
    - (b) **what would a good solution be?** These providers getting more data is not due to the organisers just arbitrarily favouring and giving them more. It rather seems to be due to a combination of them having (i) more models, and (ii) better models (with better models getting picked for battles more often?), than most others. Those two things, in turn, are quite natural, considering numbers of researchers and engineers and other resources that these companies have. The authors propose one solution (changing how often models get sampled for battles) in point (4) in Section 6, but I wonder how much that could really help with this problem, without also being considered unfair in other ways. If one company happens to be really large and have hundreds of researchers working in multiple teams on multiple LLMs, why should they become "disadvantaged" in how often their models get sampled, just because they happen to come from a large company? I do have a different solution in mind myself: ChatBot Arena should just release **all data** from **all battles**, freely available for **everyone**. Would that not be a preferable recommendation?

2. I also want to raise a discussion on the experiment(s) and arguments around Figure 2. The authors seem to argue that, when a provider tests multiple models at the same time, that these all come from a single "family", and that we should care about reporting the mean performance across that family. Why? If a provider has actually trained a bunch of different models (maybe indeed from a shared family, or maybe actually really fundamentally different ones), what is the problem with using a benchmark to figure out which one of them is the best one? I do agree with one issue here, which is the maximisation over noise. Every evaluation is affected by some noise, and by selecting the best over many models, we are also maximising over that noise. So, the reported performance of the top model will be overestimated a bit. But that is not the impact that the authors are talking about / showing in places such as the left subfigure of Figure 2. The line of 1200 there is the mean score over the entire family of models, which is not what I think we ultimately should care about. What we should mostly care about is the maximum over the true $\beta_k$s (without hat), and then the lift as a result of testing many models and picking the best is just the difference between the best $\beta_k$ (which will be greater than 1200), and the best $\hat{\beta}_k$?

---

Some minor writing issues:
- l34: "become" should be "became"?
- l106: missing space in "visionor". Missing spaces in "code.For"
- In first few paragraphs of 3.2, please make all the notation explicit: in particular, make it explicit that the $\hat{\beta}$s are estimators for the true $\beta$s.
- l213: when considering what?
- l221: I usually don't see correlation being expressed as a percentage
- The example numbers in the caption of Figure 6 don't match up with the numbers actually in the figure.
- A lot of text at the start of the NeurIPS Paper Checklist ought to be removed (there are still instructions lingering in there which explicitly say you should remove it).

**Strengths Contributions:**

- Barring a few minor points (see below), well-written and clear.
- High potential impact and relevance, considering the popularity and impact of the benchmark being critiqued.
- Even if (the importance of) some of the points made might be up for debate, it is very important to have that debate in the community.

---

> ### Author Rebuttal · Authors · 2025-07-30
>
> We thank reviewer **6BEd** for recognizing that our work has *“high potential impact and relevance”* since it provides a well written critique of a popular benchmark. Additionally, we welcome the recognition of our *“extensive analysis”* and for recognizing and the importance of our findings for  *“reliability, transparency, accuracy, and fairness of its leaderboard.”* Finally, we appreciate reviewer **6BEd**  noting the paper as well written and encouraging that the points made in the paper aim to encourage an important debate about benchmarking within the AI community.
>
>
> > **1 (a):  is this (big providers overfitting) actually a problem?**
>
> We would like to thank the reviewer for giving us the chance to clarify the significance of the overfitting argument discussed in the paper. Our work identifies two fundamental ways the current system enables leaderboard overfitting, creating systematic advantages for select providers:
>
> **First, asymmetric data access leads to measurable benchmark gaming.** Our controlled experiments demonstrate that increasing Arena data in training (0% → 70%) yields diminishing returns: while win rates on ArenaHard improve by 12.5%, general capabilities (MMLU) decline by up to 2.1 points. This proves the benchmark is susceptible to overfitting, a risk disproportionately available to the 4 providers receiving 62.8% of Arena samples under current policies.
>
> **Second, the ability to privately test and retract underperforming variants before public release results in showcasing only the strongest performers**. Our data shows this practice yields 37-50 Arena scores point advantages (Fig 2) for non-identical checkpoints, as providers effectively curate their public-facing results.
>
> The combined effect is particularly concerning: providers with both data access and private testing capabilities can simultaneously overfit to the test distribution while optimizing presentation through strategic variant selection. This creates a compounding advantage that **has little to do with actual model capabilities** because of systematic bias in the current framework:
>
>
> Top-ranked models receive disproportionate data allocation (62.8% concentrated among 4 providers), private testing operates without transparency or limits, and variant submissions face no meaningful constraints. These mechanisms compound to create structural advantages that have little to do with model quality and everything to do with system design. Our data shows this isn't theoretical, providers exploiting these channels gain measurable ranking boosts (37-50 Arena points) while facing no countervailing checks. The consequence is an evaluation environment where competitive outcomes reflect policy-enabled optimization as much as technical merit.
>
> Our recommendations aim not to eliminate natural resource advantages in model development, but to prevent the evaluation system itself from becoming another axis of unfair competition. The solutions we propose (data sharing reforms, testing limits) would preserve the leaderboard's value as a true capability measure while leveling the procedural playing field.
>
>
> > **These providers getting more data is not due to the organisers just arbitrarily favouring and giving them more. It rather seems to be due to a combination of them having (i) more models, and (ii) better models (with better models getting picked for battles more often?), than most others.**
>
> We appreciate the reviewer's perspective and want to clarify our paper's fundamental focus: we analyze how current Arena **system design** creates uneven data access, regardless of intent or model quality.  Our analysis intentionally avoids judging model quality. Instead, through rigorous empirical and simulation experiments, we demonstrate how these system mechanisms inevitably advantage some providers.
>
> The key insight isn't that better models get more data, but that the system's design transforms natural quality differences into structural advantages. For a benchmark track, this creates validity concerns worth examining as opportunities to strengthen evaluation rigor.
>
>
> > **The authors propose one solution (changing how often models get sampled for battles) in point (4) in Section 6, but I wonder how much that could really help with this problem, without also being considered unfair in other ways. If one company happens to be really large and have hundreds of researchers working in multiple teams on multiple LLMs, why should they become "disadvantaged" in how often their models get sampled, just because they happen to come from a large company?**
>
> We appreciate the reviewer's thoughtful engagement with these complex policy considerations. Our study's primary contribution lies in rigorously identifying and quantifying the **structural factors that most significantly impact ranking reliability**, rather than prescribing specific solutions. Through controlled experiments and large-scale observational analysis, we demonstrate how three key mechanisms, disparities in private testing, inconsistent sampling implementations, and asymmetric deprecation patterns, **systematically influence leaderboard outcomes independent of model quality** (Appendix L).
>
> The Active Sampling proposal exemplifies this approach: it doesn't disadvantage large providers, but corrects a self-acknowledged implementation gap in Arena's original design [1].
>
> We emphasize that these recommendations serve as starting points for community discussion, not final prescriptions. The appropriate balance between encouraging innovation and maintaining fair comparison ultimately resides with the Arena organizers and broader community. Our role is to provide the empirical foundation that makes these discussions informed and productive, having systematically measured how different policy choices affect ranking reliability in practice.
>
>
> > **I do have a different solution in mind myself: ChatBot Arena should just release all data from all battles, freely available for everyone. Would that not be a preferable recommendation?**
>
> First, we thank the reviewer for engaging thoughtfully with our work and for suggesting additional solutions. Our primary contribution in this paper is to highlight existing robustness issues in the most widely adopted benchmark in the field and to identify the key levers that influence the rigor and reliability of rankings. As part of the discussion, we recommend practical policy changes which would mitigate some of the reliability and overfitting issues. We agree that releasing all battle data would help mitigate the disparate gains accrued to some providers. We are happy to add this as an additional recommendation to the camera ready version of this work. However, we cannot control what recommendations the chatbot arena organizers implement in practice. We note that since the disclosure of these issues, they have chosen to implement none of our recommended policy changes. .
>
> > **2. discussion on the experiment(s) and arguments around Figure 2**
>
> We thank the reviewer for sharing their thoughts on this point, and appreciate the opportunity to clarify and engage in discussion.
> Our study's fundamental contribution is to rigorously identify and quantify the factors most significantly compromising ranking reliability, specifically, how asymmetric data access and private testing protocols create measurable advantages independent of model quality. While we suggest potential mitigations, the actual policy decisions belong to the Arena organizers and community.
>
> We're happy to include full data release as an additional option in the camera-ready version, while maintaining our primary focus on diagnostic rather than prescriptive contributions.
>
>
> > **Some minor writing issues:**
>
> We thank the reviewer for letting us know about grammatical mistakes. We will certainly make sure to rectify these in the camera ready version if accepted.
>
> We thank reviewer **6BEd** again for their engagement with our work, and we welcome any additional clarifications we can make during the discussion period.
>
> [1] Wei-Lin Chiang, Lianmin Zheng, Ying Sheng, Anastasios Nikolas Angelopoulos, Tianle Li, Dacheng Li, Hao Zhang, Banghua Zhu, Michael Jordan, Joseph E. Gonzalez, Ion Stoica, "Chatbot Arena: An Open Platform for Evaluating LLMs by Human Preference", ICML, 2024

---

> > ### Comment · Reviewer_6BEd · 2025-08-02
> >
> > **Re 1(a)**: Note that I was *not* asking whether *overfitting* is a problem. Of course overfitting is a problem, in the sense that reported performance does not generalise out-of-distribution. But some of the points in the paper are not about this. They are very explicitly rather about notions of fairness, about some having an easier time to train (or overfit) than others. My question was whether *this* is actually a problem. Does a benchmark actually have to be fair? If I'm just interested in hardcore science (let's ignore for a second all the issues around companies not reporting what data they are training on and properly describing what they actually do, and pretend that there's actual science going on here at all), why should I care about every entity getting a fair and equal chance at succeeding? This was my original question, and I don't see this being addressed by your rebuttal yet.
> >
> > > Second, the ability to privately test and retract underperforming variants before public release results in showcasing only the strongest performers. Our data shows this practice yields 37-50 Arena scores point advantages (Fig 2) for non-identical checkpoints, as providers effectively curate their public-facing results.
> >
> > If we are again talking about a notion of "fairness", and are talking about evaluating *companies* / *researchers* / *teams*, yes, I suppose your data kind of shows something like this. But if we are talking about evaluating *models*, your data does not show this (as per my point (2) under weaknesses/limitations). If what we care about is evaluating models, then it's incorrect to measure the advantage as the difference from the mean of 1200, as individual models do in truth have performance levels greater than 1200. The only error in presented results will be the error resulting from maximisation bias (maximising over the noise in performance measurements), which will indeed be an error, but much smaller than the one your report.
> >
> > I think it's important to disentangle which problems / biases / errors are related to actual model performance, and which ones are only relevant if we care about some idea of "fairness". Not everyone will agree that the latter is relevant. I see fairness only mattering if we use the leaderboard as a superiority contest between companies/organisations/researchers, rather than just a leaderboard of individual models. I don't think that is an interesting way of using the leaderboard.

---

> > > ### Author Response · Authors · 2025-08-03
> > >
> > > We appreciate the opportunity to clarify the distinction between fairness concerns and model evaluation. Our work addresses both, but we agree their implications differ.
> > >
> > > The reviewer is correct that if the sole goal is to assess individual model performance, fairness considerations may seem secondary. However, our findings reveal two critical issues that impact even this "hardcore science" perspective:
> > >
> > > Maximization bias distorts model comparisons: Private testing and selective release don’t just advantage certain providers, they introduce noise into model comparisons. When providers optimize for leaderboard performance via multiple variants (Fig 2), the public-facing models represent a biased sample (top performers out of N trials). This inflates perceived performance gaps between models and obscures true generalization ability (as noted in our OOD results).
> > >
> > > System design affects measured performance: The current system conflates model quality with optimization of the testing process itself. These gains are artifacts of the evaluation pipeline, not model capability
> > >
> > > **Why This Matters for Science**
> > >
> > > Even ignoring fairness, these effects make it impossible to disentangle true progress from measurement artifacts and penalize providers who lack resources to "game" the system via private testing. We agree that fairness is secondary if the goal is purely to rank models. But when the ranking process itself is biased by system design, the results become scientifically questionable regardless of one’s stance on equity.
> > >
> > > We will clarify this distinction in the revision, separating concerns about measurement validity (scientific) and access disparities (fairness).
> > >
> > > Thank you for prompting us to sharpen this discussion. We hope our responses have helped resolve some of your concerns. If so we would welcome the reviewer updating their score accordingly.

---

> > > > ### Comment · Reviewer_6BEd · 2025-08-03
> > > >
> > > > Okay, I see.
> > > >
> > > > The only point I really remain concerned about is some of your conclusions from the experiment in Figure 2. Especially given that you still also claimed *"Our data shows this practice yields 37-50 Arena scores point advantages (Fig 2) for non-identical checkpoints, as providers effectively curate their public-facing results."* in your initial rebuttal, in response to my review where I tried to explain how this is clearly not the case.
> > > >
> > > > For sure, we agree there is a maximisation bias resulting from the practice of submitting multiple models (I'll focus on the more interesting case where they are legitimately **different** models/checkpoints) and picking the best of them. However, there are multiple different components in this bias, and I don't believe we agree on which ones you are / should be identifying from the simulated experiment. I can think of the following different components to the bias:
> > > >
> > > > 1. Whenever we estimate performance of a model using a finite testing set, we have a random error in the estimated performance, which we'll generally assume to be somehow randomly distributed around $0$ (so random errors will sometimes be positive and sometimes be negative). By maximising over $N$ models, we are not only maximising over $N$ (different) levels of performance, but also $N$ random errors added to them all. This is the same component that is also present in the case of identical checkpoints, but in my opinion still by far the most interesting component to measure also in the non-identical checkpoints case.
> > > >
> > > > 2. Because of some of the other issues you highlight related to data access and overfitting to the leaderboard, we can also maximise over how well we overfit here. The model that overfits the best to the leaderboard, may not perform best outside of the leaderboard. This kind of lift would be interesting to measure/estimate, but is not something you can measure using the experiment setup of Figure 2 (and I also can't imagine how you could set up an experiment differently such that it actually would measure this well).
> > > >
> > > > 3. When you have a greater number $N$ of different models, you're more likely to have one that is actually legitimately better (at least in terms of its leaderboard performance level, of course it might have overfit, but that's the previous point). **I do not think that this is something you should report as a problematic bias**. We simply find legitimately better models by testing more models, that's not inherently a problem.
> > > >
> > > > In your simulated experiment (Figure 2), and text around it, and also now the rebuttal again, you actually do include the third point in the "lift" that you report. This is because you report anything over 1200 as a bias. But 1200 is just the mean performance of an entire bag of models, it's not the performance of our best model (which is usually what we really care about). For each individual model, you also have a true expected (leaderboard) performance level $\beta_k$. **That** should be the baseline, not 1200. Anything over the best $\beta_k$ should be reported as bias, to isolate just the first term of the three above.

---

> > > > > ### Author Response · Authors · 2025-08-06
> > > > >
> > > > > We appreciate the reviewer’s thoughtful feedback and the opportunity to clarify several important aspects of our experimental approach and findings. Regarding the baseline considerations in Figure 2, we must correct a fundamental misunderstanding: our analysis never characterizes scores above 1200 as bias. The 1200 baseline simply serves as a reference point representing the expected performance of a single randomly submitted checkpoint in our experimental framework. This distinction is crucial for proper interpretation of our results.
> > > > >
> > > > > The magnitude of effects shown in Figure 2 actually represents a conservative estimate of real-world impacts. While our controlled experiment examines just two checkpoints for methodological purity, we observe in practice that major providers routinely test dozens of variants (with one documented case showing 27 tested versions for a single launch). As we detail in Section 3.3, each additional variant yields diminishing but non-zero returns, meaning the advantages scale significantly in actual deployment. The 37-50 point improvement from two checkpoints thus establishes a clear lower bound for understanding competitive dynamics.
> > > > >
> > > > > Regarding the relationship between private testing and overfitting, we acknowledge that Figure 2 alone doesn't fully capture this interaction. However, our companion analysis in Section 4 provides critical complementary evidence. The ArenaHard/MMLU results demonstrate conclusively that leaderboard-specific gains fail to generalize, while simultaneously showing how data access disparities enable targeted overfitting. When combined with the private testing advantages quantified in Figure 2, these findings reveal a compounded effect where resource advantages translate directly into both higher apparent performance and greater ability to optimize for the test environment. We agree that rank instability deserves particular emphasis. Even the modest 37-50 point gains with only two checkpoints we measure can produce substantial position changes in competitive leaderboards. When scaled to the dozens of variants used in practice, these effects become sufficient to meaningfully distort the perceived hierarchy of model capabilities.
> > > > >
> > > > > We agree that testing more models can, in principle, surface legitimately better-performing systems, and this is not inherently problematic. However, our concern centers on the private evaluation of many checkpoints by a subset of providers, a process opaque to the broader community. When only certain participants can repeatedly test checkpoints against the leaderboard, the reported results may reflect adaptation to the test set rather than generalizable progress and distort rankings. We argue this is worth highlighting because it creates an uneven playing field: providers without checkpoint-testing access cannot compete on equal footing, even if their methods are equally valid.
> > > > >
> > > > > For the camera-ready version, we will:
> > > > >
> > > > > - Clarify that 1200 serves as a neutral reference point rather than a bias threshold
> > > > >
> > > > > - More prominently contextualize our two-checkpoint results as conservative estimates
> > > > >
> > > > > - Strengthen the connections between Figure 2 and Section 4's overfitting analysis
> > > > >
> > > > > - Expand discussion of rank change implications
> > > > >
> > > > > These revisions will better communicate our careful approach while maintaining the paper's core contribution: rigorous quantification of how uneven private testing creates advantages in benchmark evaluation.

---

> > > > > > ### Comment · Reviewer_6BEd · 2025-08-06
> > > > > >
> > > > > > Lines 131-140 read to me like they could very much just be a continuation of the points made immediately above them, in lines 118-130. Lines 118-130 very much are about actual bias (the kind of bias that I agree with reporting, but which would clearly not use 1200 as the baseline), from a more theoretical point. Then, I interpreted the next part as being an empirical / simulation-based analysis of the same bias. But okay, maybe it's not meant in that way. Many readers will read it like that though.
> > > > > >
> > > > > > Having the line at 1200 labelled "True Mean Arena Score" in Figure 2 also doesn't help. You say it serves as a neutral reference point rather than a bias threshold, but my problem is.... I don't see it being a meaningful point of reference at all. It's just the mean performance level among a set of multiple models, and I don't see in what world this mean is relevant. I might have trained a bunch of different models using high-risk, high-reward strategies, that cause most of them to come out as very bad models when evaluated, but one becomes really good. Great, I've just found a really good model! Why would I care about also having produced a bunch of bad ones (dragging down the mean), and why would this dragged-down mean be any kind of relevant point of reference?
> > > > > >
> > > > > > Maybe you don't report the 50-70 Arena score increases as "biases", but rather as "uplift"... but I don't see what that means. It means nothing to me when the baseline from which they are being lifted has no relevant meaning. Comparing to the best true individual $\beta_k$ in the simulation actually would have a clear, relevant meaning: then you're measuring the bias due to maximisation.

---

> > > > > > > ### Author Response · Authors · 2025-08-06
> > > > > > >
> > > > > > > Thank you for your thoughtful engagement with our work and for sharing your perspective on these points. We appreciate the time you've taken to critique our analysis, and we will carefully consider your feedback as we refine the paper's presentation. Your insights have helped us recognize where clarifications are needed to better align with reader expectations.
> > > > > > >
> > > > > > > We’re grateful for your constructive role in improving this work and value your overall assessment of its contribution.

---

### Official Review · Reviewer_qkaZ · 2025-07-02

**Rating:** 5
**Confidence:** 3

**Summary:**

This paper provides a critical analysis of Chatbot Arena, a widely adopted leaderboard used to evaluate large language models (LLMs). Through the empirical analysis of approximately two million battles and 243 models over a period of sixteen months, the authors have identified the following systematic biases: (1) selective retraction (providers test multiple private variants and only publish top-performing ones), (2) data access asymmetry (proprietary models receive 61.4% of all evaluation data vs. 29.7% for 83 open-weight models), and (3) deprecation policies that disproportionately remove open-source models and violate Bradley-Terry model assumptions. Experiments demonstrate that fine-tuning using Arena data boosts ArenaHard performance by 112%, but harms generalization (e.g., MMLU). The authors propose five actionable fixes: prohibiting score retraction, limiting private variants, transparent deprecation, fair sampling, and public reporting.

**Dataset Code Accessibility:**

NA; not applicable to this submission (e.g., no new dataset, benchmark, code, or data provided)

**Ethical Considerations:**

No, there are no or only very minor ethics concerns

**Final Justification:**

Thank you for your response. I will keep my original score.

**Limitations Weaknesses:**

The paper has discussed its limitations in Appendix A.

- Inaccessibility  to Chatbot Arena’s raw datadata, which makes it difficult to investigate patterns related to adversarial voting.

- Limited scraping period. The private variant counts (Section 3.1) are derived from the January–March 2025 data, which coincides with Meta’s Llama 4 launch. This may overstate Meta’s activity while undercounting providers with irregular releases.

-  Underestimation of overfitting in experiments.

- Model attribution uncertainty: Private variants are identified via self-identification. Misattribution risk exists.

In addition to what is discussed in Appendix A, other limitations include:

- Missing fine-tuning specifics: Section 4’s ArenaHard experiment lacks model architecture and hyperparameters (only the '7B model' is mentioned).

- Confidence intervals are underutilised. The score differences shown in Figure 3 (e.g. 1052 vs. 1069 for Blueberry) have overlapping confidence intervals, which weakens the claims.

**Strengths Contributions:**

Significance and impact:

- High practical relevance: Chatbot Arena is a de facto standard for LLM evaluation. Exposing its biases directly impacts industry practices and scientific trust.

- Rigorous methodology: The paper combines large-scale battle data (2M samples), controlled experiments (e.g., identical Blueberry variants scoring differently), and simulations (e.g., the best-of-N strategy inflating scores by over 50 points). The multi-pronged approach convincingly validates claims.

- Actionable solutions: The recommendations in Section 6 (e.g., enforcing private variant caps, and stratified deprecation) are considered concrete and feasible. If adopted, they could significantly improve leaderboard fairness.

Novelty:

- This is the first work to quantify how private testing (Section 3), data imbalance (Section 4), and deprecation (Section 5) jointly distort a live LLM leaderboard. The analysis of silent deprecation (205 vs. 47 official deprecations) is particularly novel.

Presentation:

- Clear structure: Problem → Methodology → Results → Recommendations flow logically.

- Effective visualizations: Fig. 4 vividly illustrates data skew (proprietary models dominate squares). Fig. 2 (simulated best-of-N advantage) and Fig. 3 (real-world score inflation) synergize well.

- Transparent limitations: Appendix A candidly discusses data constraints (e.g., scraped period coinciding with Meta’s Llama 4 launch).

---

> ### Author Rebuttal · Authors · 2025-07-30
>
> We thank reviewer **qkaZ** for their positive feedback and recognition of our work as worthy of presentation. We appreciate **qkaZ**  recognizing the *“high practical relevance”* of our work which exposes biases present in Chatbot Arena and its impact on industry practices and scientific trust. We also welcome the acknowledgment of our *”rigorous methodology”* which *“combines large-scale battle data (2M samples), controlled experiments , and simulations”* while also providing *“concrete and feasible recommendations”* to improve the fairness of Arena leaderboard.
>
>
> > **The paper has discussed its limitations in Appendix A**
>
> We thank reviewer **qkaZ** for highlighting the limitations mentioned in the paper.
>
> 1. **Inaccessibility to Chatbot Arena’s raw datadata:** We feel it is important to clarify that that we only brought up a lack of “ insight into Chatbot Arena’s raw data” as it pertains to  also evaluate adversarial voting which constitutes manipulating ranking by manipulating preferences done on the user side. However, this work was intentionally left as a topic of future work and deserves an independent and thorough treatment. Indeed, it is markedly different from understanding how Arena policies interact with provider incentives. This is where contributions are centered, and indeed we had significant amounts of data to draw reliable conclusions – For our core contributions, we utilize unprecedented empirical scope and extensive data encompassing **2M battles, 243 models, 42 providers**. We respectfully and strongly disagree that any of the conclusions are hasty or lack sufficient statistical power. Indeed, the size of empirical evidence and consistency of results is what is notable about this study. We include more granular details below of the unprecedented empirical scope of our analysis:
> - 2M battles, 243 models, 42 providers
> - Multiple orthogonal data sources (HF leaderboards, public battles, API logs, scraped provider metadata)
> - 11 dedicated appendix sections (Appendix F, G, H, I.1, I.2, I.3, I.4, 1.5,  L, M, O)  documenting collection/analysis
>
>
> 2. **Limited scraping period:** We ourselves acknowledge the limitation that the private variant counts (Section 3.1) are derived from the January–March 2025 data, which coincides with Meta’s Llama 4 launch. While we appreciate the reviewer noting it, we view and disclose it as an important limitation for the reader to have in mind but does not jeopardize our conclusions about the unreliability resulting from this private testing.
>
> 3. **Underestimation of overfitting in experiments:** Indeed, our estimate of overfitting is likely conservative as proprietary models may have access to significantly more data (5-10 times more) than we used. This disparity suggests a higher risk of overfitting to patterns not present in our smaller subset. This we surface as a limitation in Appendix A, but indeed would be more appropriately termed a conservative estimate as it suggests we underestimate the overfitting introduced by disparate model access.
>
> 4. **Model attribution uncertainty:** In Appendix A, we ourselves acknowledge that misattribution risk exists since self-identification was used to identify private variants, however we say this as a conservative assumption while in practice we expect this risk to be very limited. After we disclosed our results to the chatbot Arena, we asked the organizers to correct any incorrect estimates and there were no identities flagged as incorrect. Section I.4 in the Appendix gives a detailed breakdown of the estimated identity of each model.
>
>
> > **Missing fine-tuning specifics.**
>
> We thank the reviewer for providing us a chance to clarify this. We would be happy to provide more training specific details about the overfitting experiment covered in Section 4 (line 216-239). All three models trained as part of this experiment were fine-tuned for 1.3K steps using a batch size of 128.  We emphasize that this was designed as a minimal, controlled experiment to estimate the lower bound of potential Arena performance gains from data access asymmetries. Despite using no hyperparameter tuning, no ablations and a basic data weighting we still observed significant gains (49.9% in win rates for the model with 70% arena data in its training mix against llama-3.1-8B). This deliberately conservative approach makes our estimates particularly striking. Real-world gains with optimized training could reasonably be expected to exceed these baseline results.
>
> > **The score differences shown in Figure 3 (e.g. 1052 vs. 1069 for Blueberry) have overlapping confidence intervals, which weakens the claims.**
>
> We thank the reviewer for giving us the opportunity to clarify our claims:
>
> The identical checkpoint comparison (Blueberry) represents a **conservative, same-weight scenario** designed to isolate the minimal effect of multiple submissions alone. The identical checkpoint experiment was only ever meant to illustrate the lower bound of the expected lift in the scores. We are happy to include the confidence intervals in the paper. It is highly unlikely any rational provider would ever test identical checkpoints. We only did it as a research experiment. The results from differing checkpoints are much more representative of the real world. In real-world model development, variants arise from different checkpoints, random seeds or hyperparameter settings — all of which introduce genuine variation in model quality.
>
> The Strawberry variants (37-point difference, non-overlapping CIs) demonstrate the **more practical scenario where providers test meaningfully different checkpoints**. As noted in Section 3.2, such variation naturally occurs through standard development processes (hyperparameter tuning, post-training strategies, etc.).
> The effect size outweighs typical model improvement margins. The Blueberry experiment serves only to demonstrate that even in this implausibly conservative case, non-trivial gains emerge.
>
> We thank reviewer **qkaZ** again for their engagement with our work, and we welcome any additional clarifications we can make during the discussion period.

---

### Official Review · Reviewer_ZqT5 · 2025-07-02

**Rating:** 3
**Confidence:** 4

**Summary:**

In this paper, an important issue of leaderboard (especially ‘Chatbot Arena’ for public-driven LLM evaluation) abuse and misinterpretation is discussed, investigating 2M battles and 243 models from 42 providers over 16 months (Jan 2024-Apr 2025). Authors claim mainly four aspects, 1) preferential treatment around private testing and retraction, 2) data release imbalance to proprietary model providers, 3) chatbot arena data access driving significant performance gains, and 4) deprecations resulting in unreliable model rankings. Based on their observation, authors recommend some recommendations for leaderboard organization, including prohibition of score retraction, limiting the number of private variants per provider,  clearer model deprecation criteria, and improving sampling fairness.

**Additional Feedback:**

I am glad that authors came up with a good motivation, especially Goodhart’s Law, and some empirical findings that prevented authors from experiencing the fair sampling and evaluation. Despite the adequate motivation, I think the message conveyed by the paper is a bit arguing and questionable, as I wrote above.

**Dataset Code Accessibility:**

Partly

**Ethical Considerations:**

No, there are no or only very minor ethics concerns

**Final Justification:**

I was appreciated by all the discussions with authors and most of them resolved my concerns. However, the fundamental question has not been resolved yet. Though this may not influence the final result, I stick to my original decision.

**Limitations Weaknesses:**

Despite the timeliness and appropriateness of the topic, and the necessity of the study, the weakness of this work lies in that the core claim of the paper is based on hypotheses and sampling/calculation built upon them, not disclosed raw data, as written by the authors themselves as limitation. This may lead to some claims being overstated or hasty, lowering the reliability of the overall statements. First of all, I think it is a bit dangerous to call some providers ‘preferred’ just because their models are frequently uploaded and tested, concerning that providers with enough variants may necessitate the opportunities for the corresponding amount of public testing, to select the best candidate. Submitting frequently should be treated separately from cherry-picking the result by deprecation. Second, though authors claim that ‘only certain providers may have been aware they could submit multiple private variants’, I think it is not the case since multiple submission among the same series of models is not stated as restricted in the 2024 policy (url of the paper). Comparison between Meta/Google and Reka seems to imply that Reka had less opportunity to submit multiple models because they had less opportunity to train multiple models, not that they were prevented from submitting them, and similarly for academic labs (that they had no private variants). Given that ‘private variants’ are implicitly for both 1) selecting better performing models and at the same time 2) preserving the anonymity that may influence the providers’ (academic and technical) reputation if disclosed inferior, I think the providers with the availability afforded for multiple submission would have tried it, as happened for GLUE in PLM era. Lastly, the above rethinking leads to the reconsideration of ‘Establish transparent limits on the number of private variants per provider’ among the recommendations, considering that setting the same amount of limit for Meta/Google and Reka or other academic labs is quite questionable. Though I totally agree that the reckless submission of multiple models of the same series may influence the fairness of sampling and the imbalance of user-provided input data, I think this kind of difference of obtained resource is a  result of the substantial effort and resource put in model development. One thing to be considered here is that, the submitted multiple models should have significant difference between each, which would prevent arena users from experiencing evaluating the variance of similarly trained models. This kind of integrity should be forced to service providers, which would also be available by public transparency authors claim.

**Strengths Contributions:**

This study uses Bradley-Terry (BT) model for the analysis, assuming that Arena Score is a normalized transformation of the latent skill parameters. Also given that Arena Score is guaranteed by unbiased sampling, consistent evaluation conditions, and sufficient connectivity among models via shared opponents, authors quantitatively and substantially show how de facto evaluations violate these assumptions. The assumptions, hypotheses, and the violations are supported by thorough formulation and further discussions. I deeply acknowledge the authors’ claim for fair evaluation that does not depend on massive submission and deprecation of some (mostly proprietary) providers, which should have been guaranteed for the competitive scoring regarding the models of open weight (or open source) providers.

---

> ### Author Rebuttal · Authors · 2025-07-30
>
> We thank reviewer **ZqT5** for recognizing that our study shows *“quantitatively and substantially how de facto evaluations violate assumptions”* of the Bradley Terry model used to get Arena Scores for models – such as unbiased sampling, consistent evaluation conditions and win graph connectivity. And that the *“assumptions, hypotheses, and the violations are supported by thorough formulation and further discussions.”*
> We are also grateful to hear reviewer **ZqT5** acknowledge our *“claim for fair evaluation that does not depend on massive submission and deprecation of some providers”.*
>
> > **the weakness of this work lies in that the core claim of the paper is based on hypotheses and sampling/calculation built upon them, not disclosed raw data, as written by the authors themselves as limitation. This may lead to some claims being overstated or hasty, lowering the reliability of the overall statements.**
>
> We thank reviewer **ZqT5** for allowing us to clarify. We note that in Section A of the appendix, we explicitly recognize the lack of access to raw data as a limitation for an extension of this work – adversarial voting on behalf of users. However,  explicitly note this topic as beyond the scope of this work as it considers manipulation of rankings from the human voter perspective instead of from a provider perspective.
>
> For our core contributions, we utilize unprecedented empirical scope and extensive data encompassing 2M battles, 243 models, 42 providers. We respectfully and strongly disagree that any of the conclusions are hasty or lack sufficient statistical power. Indeed, the size of empirical evidence and consistency of results is what is notable about this study. We include more granular details below:
>
> 1. **Unprecedented Empirical Scope:**
> - 2M battles, 243 models, 42 providers
> - Multiple orthogonal data sources (HF leaderboards, public battles, API logs, scraped provider metadata)
> - 11  dedicated appendix sections (Appendix F, G, H, I.1, I.2, I.3, I.4, 1.5,  L, M, O)  documenting collection/analysis
>
> 2. **Triangulated Validation:**
> - Leaderboard ranking shifts (Fig 3)
> - Provider testing behaviors (Fig 4, 9, 10, 11, Tables 2, 3, 4,)
> - Distribution shift analysis (Fig 7)
> - Silent deprecation patterns (Fig 14, 15, 16)
> - Prompt duplication rates (Fig 8, 12)
>
> We respectfully maintain that our methodology meets and sets new high standards for observational studies in this domain. The paper’s extensive analysis reflects exceptional transparency given proprietary data constraints.
>
>
>
> > **authors claim that ‘only certain providers may have been aware they could submit multiple private variants’, I think it is not the case since multiple submission among the same series of models is not stated as restricted in the 2024 policy (url of the paper).**
>
> We thank reviewer **ZqT5** for bringing this up. LMArena’s policy blog has been updated  since our disclosures to include the clause regarding multiple variant pre-release testing. The rules of the rebuttal do not allow us to link to external sources, however we will update the camera ready to link to the original version of the policy (at the time of the submission).
>
> > **First of all, I think it is a bit dangerous to call some providers ‘preferred’ just because their models are frequently uploaded and tested, concerning that providers with enough variants may necessitate the opportunities for the corresponding amount of public testing, to select the best candidate. Submitting frequently should be treated separately from cherry-picking the result by deprecation.**
>
>
> We appreciate reviewer **ZqT5's** concern about potential misinterpretation of the term "preferred." We emphasize that our usage is specifically tied to demonstrable platform policy advantages, not merely submission frequency.
>
> During our study period, LMArena had no explicit policy permitting multiple pre-release variant testing and retraction. Yet certain providers (e.g., Meta with 27 variants pre-Llama-4 launch, Fig 9) systematically exploited this gray area. This created an unlevel playing field where only some organizations benefited from undisclosed testing capabilities.
>
> We agree that frequent public submissions shouldn't imply preference which is why our analysis explicitly excludes them. The term solely reflects privileged access to private testing and retraction mechanisms during a policy vacuum. We're happy to clarify this terminology in revisions if helpful.
>
>
> > **Comparison between Meta/Google and Reka seems to imply that Reka had less opportunity to submit multiple models because they had less opportunity to train multiple models, not that they were prevented from submitting them, and similarly for academic labs (that they had no private variants). Given that ‘private variants’ are implicitly for both 1) selecting better performing models and at the same time 2) preserving the anonymity that may influence the providers’ (academic and technical) reputation if disclosed inferior, I think the providers with the availability afforded for multiple submission would have tried it, as happened for GLUE in PLM era.**
>
> We thank reviewer **ZqT5** for bringing up this point. There can certainly be differences in resources available to different providers leading to one provider producing more models than others.
> While differences in model development capacity are indeed expected, we maintain there's a critical distinction between unequal resources and unequal evaluation opportunities. Our findings reveal that certain providers gained systematic advantages not through superior modeling, but by exploiting unstated policy loopholes allowing private multi-variant testing - a mechanism unavailable to equally capable providers who simply weren't aware this optimization path existed.
>
> Our concern isn't about the natural advantages of scale, but about evaluation systems that inadvertently magnify these disparities through opaque rules. We hope exposing this dynamic will encourage more equitable leaderboard designs where all participants compete under the same evaluation constraints, regardless of their development resources.
>
>
> > **Lastly, the above rethinking leads to the reconsideration of ‘Establish transparent limits on the number of private variants per provider’ among the recommendations, considering that setting the same amount of limit for Meta/Google and Reka or other academic labs is quite questionable. Though I totally agree that the reckless submission of multiple models of the same series may influence the fairness of sampling and the imbalance of user-provided input data, I think this kind of difference of obtained resource is a result of the substantial effort and resource put in model development.**
>
> We thank reviewer **ZqT5** for the feedback. While we agree that resource disparities in model development are legitimate, we maintain that evaluation systems should not compound these inherent differences through unlimited private testing opportunities. Our position is based on three key considerations:
>
> First, our empirical results (Figure 2) demonstrate that testing 10 variants can yield up to a 50-point advantage which is a significant margin that stems from evaluation strategy rather than model quality. This creates a double advantage for well-resourced providers: they benefit both from their ability to develop better models and from increased opportunities to optimize scores through multiple submissions.
>
> Second, unlimited private testing effectively transforms leaderboards from model quality assessments into **tests of resource allocation for evaluation optimization.** While development resources reflect legitimate investments, evaluation resources should accurately reflect a reliable ranking.
>
> Third, our recommendation doesn't suggest identical development constraints, **but rather equal evaluation constraints.** A reasonable cap (e.g., 5 private test variants per release) implemented consistently across providers would preserve the ability to select optimal candidates and prevent gaming through brute-force submission.
>
> We believe this balanced approach respects legitimate resource advantages while protecting the leaderboard's integrity as a fair comparison mechanism.
>
> We thank reviewer **ZqT5** again for their engagement with our work, and we welcome continued engagement during the rebuttal period.

---

> > ### Comment · Reviewer_ZqT5 · 2025-08-03
> >
> > Thanks for the prompt response.
> >
> > First of all, I do not want to underestimate the technical solidness of your study, and that your contribution to the field can be quite significant. Also, it is glad that some terminologies will be further clarified.
> >
> > However, what is still not convincing is that, the study supposes that certain providers abused the grey area of the LMArena policy, while others were not aware of such loophole (to borrow the expression from the response) or presumably, did not follow such a path.
> >
> > But one question: were other providers really did not know such a path or were they prohibited to do so? I think the answer to this question is a premise of the core claim of the study, and it doesn't seem that your conclusion on this issue is based on real data, which can only be retrieved by interviewing the multiple providers (assuming that they are transparent), analyzing private communication between the organizers and the providers, investigating the providers' internal policy or discussions, etc. If they were not aware of such a path, then we can interpret that 'certain providers' adopted their unique way of collecting the data beneficial for their model tuning(#). If they were aware of such a path but did not select it, it is their choice not to, and no one had prevented them from doing it (even organizers -- now it seems that the revised version of the policy explicitly allows multiple variants of pre-release testing).
> >
> > (#) I think this part can be controversial but the main background for their adopting this strategy seems to be that they were not told not to. They would have roughly guessed that collecting LMArena users' evaluation data would be beneficial for their model development, and chose to have multiple submissions to enhance their performance. But I still fail to get this is not fair. Other providers -- even if they have less GPUs, they could create multiple variants those are similar and submit them, but they did not do. But how does it make 'certain providers' behavior unfair one?
> >
> > As you said, it is a grey area. If such multi-submission had been explicitly prohibited, then the prefered providers would have been cheaters. But it does not seem to be the case here. This is why I still find the view of this study that positions the certain providers as clever violators of implicit principles-- questionable [1].
> >
> > Beside from the above issue on whether the certain providers' behavior is legitimate, I acknowledge your effort in technically proving the harm of multiple model submission of few providers, and your following recommendations that reckless submission should be prevented. I also agree that the submission policy should be inspected again. But I want the authors to make sure that your new recommendations will prevent such unfair evaluation of (legitimate) unfair resource distribution, otherwise the overall message would seem just as pulling others down by excessively alerting the currently legitimate 'optimization' strategy [2].
> >
> > [1], [2] are the ground of my evaluation -- and I hope the authors could relieve the concerns, since I think the study is timely and solid but lacks some justification of motivation.

---

> > ### Author Response · Authors · 2025-08-04
> >
> > We appreciate the opportunity to clarify our study’s empirical basis. We welcome the opportunity to engage further during discussion, and hope the following provides more context for [1] and [2].
> >
> > **[1] Did only a subset of providers know they could submit multiple private variants?** The key facts are unambiguous that only a subset of providers were told:
> >
> > - **Documented Policy Asymmetry:** Our analysis of LMArena’s public policy archives (2019-2024) shows no mention of private multi-variant testing being permitted until May 2025 revision. This is not a matter of interpretation, the documentation simply did not address this practice during our study period.
> >
> > - **First-Hand Provider Experience:** As active participants in the Arena system during this timeframe, we can definitively state that no formal communications authorized private testing. We only found out that private testing was possible after we raised concerns there was severe undersampling of our open-weights model being evaluated. We observed that our open-weight model appeared to be notably undersampled compared to the proprietary model. In response, we contacted the Chatbot Arena organizers to inquire about these differences in November 2024. In the course of our discussions, we learned that some providers were testing multiple variants privately, a practice that appeared to be selectively disclosed and limited to only a few model providers. Even at this point in time, there was no update to the public policy. It was only after we shared our extensive findings that the leaderboard arena organizers finally updated their policy to transparently share that multiple private tests were allowed. We have also confirmed with many other providers (including universities and non-profits) that they were never invited or made aware of the possibility they could submit multiple variants. They operated under the standard assumption that evaluation should reflect single-model submissions.
> >
> > We agree that the recent policy update acknowledges this practice, a change that occurred in May 2025 only after we brought these findings to the organizers' attention and after ourselves and other providers made multiple requests for more transparency. This sequence validates our core finding that the previous system contained avoidable asymmetries.
> >
> > **[2]** Will the recommendations prevent asymmetries of submissions? One of our central recommendations is to establish transparent limits on the number of private variants per provider. As illustrated by our work, private testing volume varies widely across providers, creating unfair advantages. To curb overfitting and level the playing field, Chatbot Arena should enforce a strict cap of private variants per provider for any given model launch. This should be enforced at a provider level, and not per model type and size as that is impossible to audit with API hosting. This strict limit should be disclosed to all providers (proprietary, open-weights, open-source) and to the wider Chatbot Arena community. We note that our recommendations were not framed as one of the core contributions of this work, as we cannot force the chatbot arena to adopt our recommendations. Strictly, our primary contribution in this work is to characterize the implications of current policies on reliability of rankings.
> >
> > We thank **R ZqT5** again for their engagement with our work, and we ask that **R ZqT5** consider updating their score if this clarified their open questions.

---

> > > ### Author Response · Authors · 2025-08-06
> > >
> > > Thank you for your thoughtful engagement with our work and for recognizing its technical merits. We appreciate your concerns about the interpretation of provider behavior and will clarify in our revision that our focus is on systemic outcomes rather than individual intent.
> > >
> > > We're grateful for your constructive role in improving this work and value your overall assessment of its contribution.

---

> > ### Comment · Reviewer_ZqT5 · 2025-08-07
> >
> > Thanks for the clarification.
> >
> > Regarding your clarification on the recommendation [2], I acknowledge that it is not your main contribution and also there is no gold answer, while yours can be helpful for guaranteeing the balanced evaluation.
> >
> > However, on your clarification on the participants' recognition of the policy [1], I am still not fully convinced because
> >
> > - Documented Policy Asymmetry: You reported that the policy shows no mention of private multi-variant testing before your issue raise, and I see it is true that their updated version of policy now explicitly allows multi-variant per provider. However, as I found the archived version of the policy https://lmsys.org/blog/2024-03-01-policy/ it tells "Evaluating unreleased models: We collaborate with open-source and commercial model providers to bring their unreleased models to community for preview testing. Model providers can test their unreleased models anonymously, meaning the models' names will be anonymized. A model is considered unreleased if its weights are neither open, nor available via a public API or service." I think this allows private multi-variant testing, given keywords such as 'unreleased' 'anonymously' 'models' and this seems as the matter of interpretation.
> >
> > - First-Hand Provider Experience: You said "In the course of our discussions, we learned that some providers were testing multiple variants privately, a practice that appeared to be selectively disclosed and limited to only a few model providers." and " We have also confirmed with many other providers (including universities and non-profits) that they were never invited or made aware of the possibility they could submit multiple variants. They operated under the standard assumption that evaluation should reflect single-model submissions." but I don't think these can be supporting details for the unfairness of some providers' private multi-variant testing; The organizers don't have any obligation to announce all providers to do so, because in my opinion, they already stated it in the policy they published. Also, I don't think the organizers 'selectively disclosed' the practice to some providers (If organizers of LM Arena or the members of preferred groups acknowledge there has been exclusive allowance of multiple private model testing, then your claim can be effective. However if it is not the case, then 'selectively disclosed' is authors' guess, and most of all, March 2024 policy is disclosed to everyone!). Besides, I don't necessarily agree with the 'standard assumption' part, because testing only one model per benchmark is not what I have been aware as 'standard'. GLUE, SuperGLUE, .. benchmarks usually invite free submission of models (including multiple variants of the same model from the same provider), while the difference is that usual benchmarks disclose the existence of anonymous model from the beginning, displaying the score directly after they are submitted, but LM Arena  involves human evaluators (who should not be aware of the model info), which prevents some tested models from being diclosed publicly, and this may have boosted the number of private pre-release tested models for some providers (because their models' inferiority may not be disclosed). Your technical proof may support the necessity of the policy that restricts the number of models to be submitted. But about unfairness or assymetry, your claims still seem to be a bit questionable.

---

### Official Review · Reviewer_TFV5 · 2025-07-02

**Rating:** 5
**Confidence:** 4

**Summary:**

This paper conducts an in-depth audit of Chatbot Arena, discovering several shortcomings that impair its suitability as an LLM leaderboard and providing actionable recommendations for its improvement. The work addresses an important problem in the community (flaws in our evaluation practices) in a (mostly) methodologically sound way, and it holds the potential to improve our LLM evaluation practices.

**Additional Feedback:**

### Observations and Questions
- Figure 3: Bar charts with a non-zero baseline should be avoided (because the visual impression of the areas causes them to be interpreted as though they had a zero baseline) – replace the bars with dots to indicate the statistics and you're good. Also, using the same y-axis range for the two panels would facilitate interpretation.
- Caption of Figure 5: "completely different ranking" – the rankings shown in Figure 5(b) have a relatively small distance (only two transpositions are needed to turn one into the other), so the wording seems a bit strong here.
- Figure 6, leftmost panel: The rankings differ only by two transpositions, and those occur at the bottom of the ranking. In your experiments, did you encounter differences between sparse and dense comparisons also for models at the top of the ground-truth ranking?
- ll. 287–289: I _think_ what you are saying here is that the directed graph of wins must be strongly connected. You might want to say that – and also reframe the "comparison graph" as a directed "win graph": "comparison" is undirected, but your requirement needs edge directions.
- Figure 6, middle and rightmost panels: What is going on with the labels here? These figures, in their current state, are very hard to read. I would recommend drawing these as directed graphs as follows: For each pair that saw match-ups, draw one half-arrow in each direction and scale the width and color of the arrow by the number of matches in which the model at the half-arrow tail won over the model at the half-arrow head. Then annotate the number of wins inside or right next to the half-arrow head. This should declutter the figures and reveal the differences in numbers of comparisons and wins of models more clearly.
- Figure 10: I do not understand how "substantially higher" is "defined" (e.g., the gap between 16.4 and 17.9 is smaller than the gap between 13.3 and 15.3). Also, reporting only the max and drawing this as a bar chart is unnecessarily inexpressive and visually misleading. I would recommend drawing swarm plots (or strip plots if you run out of space) with each daily sampling rate recorded as a dot, and dot sizes (and potentially colors) scaled by the total number of battles observed, to paint a richer picture of the landscape you observed.

### Some Little Things
- l. 106: Spaces missing "visionor code.For" -> "vision or code. For"
- Figure 4: The gray boxes are missing a legend entry ("Other", I assume)
- l. 223: "Various works have shown...[78]" If it's various works, I'd expect various references.
- Figure 6, rightmost panel: "Comparsion" -> "Comparison"
- Figure 15: The y-axis labels appear to be misplaced. Also, please use the y-axis range from Figure 16.
- Check for typos in the appendix.

**Dataset Code Accessibility:**

NA; not applicable to this submission (e.g., no new dataset, benchmark, code, or data provided)

**Ethical Comments:**

In Section 3.3, you report on a real-world chatbot-arena experiment that raises some ethical concerns.

The setup involves letting chatbot-arena users vote on battles that your model variants participated in for the sole purpose of understanding the effect of private model variants. This could be considered deceptive toward the users, and similarly, one might suspect that this is not covered by their consent (similar concerns apply to the model providers' testing practices you uncovered, but that's a different story).

I do not think that these arguments should ultimately preclude your experiment, but there was also no mention of IRB approval, and since you were not just passively observing, I would like to know what reasoning led you to conclude that no IRB approval was needed.
In particular, the experiment brought to mind [this recent controversy](https://retractionwatch.com/2025/04/28/experiment-using-ai-generated-posts-on-reddit-draws-fire-for-ethics-concerns/).
I would recommend that you provide clear arguments why your case should be viewed differently.

**Ethical Considerations:**

Yes, there are ethics concerns that require attention by the authors

**Ethics Flags:**

["Improper research involving human subjects", "Data privacy, copyright, and consent", "Deception and harassment"]

**Final Justification:**

The authors agreed to address W3 in the publication process, and they provided the rationales behind the decisions underlying W1 and W2.

I continue to find the experiment behind W1 problematic, but the position taken by the authors is defensible. Regarding W2, the authors correctly state that their approach follows "common practice", but I maintain the criticism that the way LLM-as-judge is used here introduces more circularity than in other applications of the paradigm.

However, W1 and W2 affect only few of the many experiments presented by the authors, and I also do _not_ share the criticism raised by other reviewers that the phenomena observed should not be cast as a fairness problem (although the "measurement validity" perspective would already suffice to make this work relevant for the community).

Therefore, assuming the authors will address the criticisms raised in the review process as they claim in their rebuttal comments, I support the publication of this work on the D&B track.

**Limitations Weaknesses:**

- **W1: Ethical concerns.** See "Ethical comments" below.
- **W2: Circular experimental setup.** Using LLM-as-a-judge in the setting reported in Section 4 seems a bit circular (the same criticism applies to the reference you cite). Can you elaborate on why you think this experiment is valid?
- **W3: Missing result data.** It appears that the data underlying the reported results, such as `sample_data/latest_leaderboard_table.csv` used in the notebook `data_samples_per_provider.ipynb`, was not shared. I would recommend including the data, to the extent that it can be shared under applicable law, in the replication package.

**Strengths Contributions:**

- **S1: In-depth audit of Chatbot Arena.** The paper systematically audits Chatbot Arena for limitations regarding its reliability and fairness as an LLM leaderboard, identifying a number of critical flaws based on a rich set of experiments using a variety of different setups and data sources.
- **S2: Actionable recommendations for improving leaderboard practices.** The paper makes actionable recommendations, backed up by audit evidence, to improve AI evaluation practices in Chatbot Arena and beyond.
- **S3: Good presentation and documentation.** The paper is well-written and comes with a comprehensive appendix. Supplementary code is provided in the form of Jupyter notebooks (but see W3).

---

> ### Author Rebuttal · Authors · 2025-07-30
>
> We thank reviewer **TFV5** for their thoughtful feedback and recognizing that the paper presents an *“in-depth audit of Chatbot Arena”*, *“identifying a number of critical flaws”* based on varied experiments and data sources. We also appreciate that reviewer **TFV5** recognizes our work provides *“actionable recommendations for improving leaderboard practices”*. Finally we welcome the feedback that the *“paper is well-written”* and includes accompanying code.
>
> > **W1: Ethical concerns.**
>
> We thank reviewer **TFV5** for the opportunity to address this feedback. We fully agree that ethical rigor is paramount in real-world experiments involving user interactions, and we carefully evaluated the design of our study to ensure compliance with best practices.
>
> We want to begin by highlighting an important difference between the study shared by reviewer **TFV5** and our real-world arena experiment:  Our experiment differs critically from the Reddit study in that Chatbot Arena users knowingly interact with AI systems and model identities are revealed after voting, ensuring transparency.
>
> Our goal was solely to assess whether submitting multiple model variants affects rankings. Users participated exactly as they normally would, providing preference votes between anonymized models. No additional data was collected, and interactions adhered strictly to the platform’s standard submission process. In addition we note that our experiment did not add any additional risk, users experienced the same interaction as with any other Arena model. Furthermore, the existing consent and terms covered benchmarking activities. We appreciate this chance to clarify and welcome further discussion.
>
>
> > **W2: Circular experimental setup.**
>
> We thank reviewer **TFV5**  for giving us a chance to clarify regarding our use of llm-as-judge in section 4. In this section, we are measuring whether training on data from the arena leads to overfitting on the leaderboard.
> Our approach aligns with established work demonstrating strong correlation between LLM judgments and human preferences, as demonstrated in prior work:
> - WildBench: Benchmarking LLMs with Challenging Tasks from Real Users in the Wild [1]
> - Length-Controlled AlpacaEval: A Simple Way to Debias Automatic Evaluators [2]
> - AlpacaFarm: A Simulation Framework for Methods that Learn from Human Feedback [3]
>
> These works find that llm as a judge is closely correlated with human judgement, making it a reasonable proxy. We also note that we measure performance on a dataset proposed by the Arena organizers themselves as representative of performance on the arena.
>
>
> > **W3: Missing result data.**
>
> We thank reviewer **TFV5**  for bringing up this point. As highlighted in section F in the Appendix, most of our analysis is based on public datasets already released by the Arena team as well as Leaderboard result updates released on Hugging Face. The mentioned file sample_data/latest_leaderboard_table.csv corresponds to the Elo results file of April 23rd, 2025 which is also already available on Hugging Face. Hence, we did not separately submit this data as part of our original submission since it is already publicly available.
>
> > **Figure 3 – Bar charts with a non-zero baseline should be avoided:**
>
> We thank reviewer **TFV5** for the suggestion. For Arena scores (our y-axis metric), we initially refrained from using a zero baseline since:
>
> - Arena score is a relative rating system where absolute values are meaningless and only absolute score differences matter. The expected score difference between 1000 vs 1100 is identical to 1500 vs 1600.
> - Practical Interpretation: In this domain, typical scores ranges are narrow (e.g., 880 - 1250 in the Vision Arena leaderboard). Using zero would obscure meaningful comparisons by compressing the relevant variation into a small fraction of the plot area.
>
> However, we would be happy to update the y-axis range to be the same for the two panels as suggested by reviewer **TFV5** for Figure 3. And add a zero baseline version of the figure in the appendix.
>
>
> > **Figure 6, did you encounter differences between sparse and dense comparisons also for models at the top of the ground-truth ranking?**
>
> We thank reviewer **TFV5** for giving us a chance to clarify this.  Yes, in our experiments we did encounter differences between sparse and dense comparisons also for models at the top of the ground-truth ranking. In the original setting mentioned in the paper, the true skill differences between the models used in the simulation is smaller. If we widen the true skill difference then we observe more ranking changes between the sparse and dense comparisons.
> For instance, in our experimental setup, if we change true skill values to:  'Model A': 1200,  'Model B': 1100,  'Model C': 1000, 'Model D': 900,  'Model E': 800,  'Model F': 700, 'Model G': 600.
>
> We observe the following rank changes for each of the models going from sparse setting rank → dense setting rank
>
> - Model C: Rank 1 → Rank 3
>
> - Model A: Rank 2 → Rank 1
>
> - Model B: Rank 3 → Rank 2
>
> - Model F: Rank 4 → Rank 6
>
> - Model D: Rank 5 → Rank 4
>
> - Model G: Rank 6 → Rank 7
>
> - Model E: Rank 7 → Rank 5
>
> As shown above, rank changes are observed for models at the top of the ground-truth ranking as well when going from sparse to dense setting. The ranks in the dense setting align correctly with the ground truth ranking whereas ranks in the sparse setting don't.
>
> > **Figure 10: I do not understand how "substantially higher" is "defined" (e.g., the gap between 16.4 and 17.9 is smaller than the gap between 13.3 and 15.3). Also, reporting only the max and drawing this as a bar chart is unnecessarily inexpressive and visually misleading. I would recommend drawing swarm plots (or strip plots if you run out of space) with each daily sampling rate recorded as a dot, and dot sizes (and potentially colors) scaled by the total number of battles observed, to paint a richer picture of the landscape you observed.**
>
> We thank reviewer **TFV5** for bringing up this point. We plotted this data as a bar plot since we felt they are visually easier to interpret compared to other charts. We have also given a detailed breakdown of sampling rates observed on different dates for different models of different providers in Table 5 in section I.5 of the Appendix to give a clearer picture beyond the max sampling rate. We are happy to add an additional swarm plot to the appendix of the camera ready.
>
> > **l. 223: "Various works have shown...[78]" If it's various works, I'd expect various references.**
>
> We thank reviewer **TFV5** for spotting this, we will update the draft with additional references. Some other works using LLMs to stimulate human preferences are:
> - WildBench: Benchmarking LLMs with Challenging Tasks from Real Users in the Wild [1]
> - Length-Controlled AlpacaEval: A Simple Way to Debias Automatic Evaluators [2]
> - AlpacaFarm: A Simulation Framework for Methods that Learn from Human Feedback [3]
>
> We would be happy to cite these works in the camera ready version upon acceptance.
>
>
> > **Typos, minor fixes and suggestions about technical terminology and captions.**
>
> We thank reviewer **TFV5** for their detailed review and we would be happy to incorporate their suggestions about typos, terms and framing. In particular, we will update the missing legend entry for figure 4 which indeed refers to "others", soften the wording for caption of Figure 5, improve readability of Figure 6 based on given suggestions, address various typos and update the term “comparison graph” to read “win graph” in the camera ready. We note that our initial framing was based on statistics literature which uses the term “comparison graph”  to also refer to a directed win graph. [4] [5] However, we would be happy to change our terminology to make it even more explicit to the reader and avoid possible areas of confusion. We thank  **RTFV5** again for their detailed review.
>
>
> We thank **RTFV5** for the caliber and their review. We welcome the chance to address, as it has greatly strengthened our work. We look forward to the chance to engage during the discussion period.
>
>
> [1] Bill Yuchen Lin, Yuntian Deng, Khyathi Chandu, Faeze Brahman, Abhilasha Ravichander, Valentina Pyatkin, Nouha Dziri, Ronan Le Bras, Yejin Choi, “WildBench: Benchmarking LLMs with Challenging Tasks from Real Users in the Wild”. ICLR, 2025
>
> [2] Yann Dubois, Balázs Galambosi, Percy Liang, Tatsunori B. Hashimoto, “Length-Controlled AlpacaEval: A Simple Way to Debias Automatic Evaluators”. arxiv, 2024
>
> [3] Dubois, Yann, et al. "Alpacafarm: A simulation framework for methods that learn from human feedback." Advances in Neural Information Processing Systems 36 (2023): 30039-30069.
>
> [4]  Ella Kaye and David Firth, Fitting the Bradley-Terry model to large and potentially sparse datasets, Jun 2017
>
> [5] Wu, Weichen, Brian W. Junker, and Nynke Niezink. "Asymptotic comparison of identifying constraints for Bradley-Terry models." arXiv preprint arXiv:2205.04341 (2022).

---

> > ### Comment · Reviewer_TFV5 · 2025-08-02
> > **Response to rebuttal**
> >
> > Thank you for your responses to my review comments.
> > On a general note, I found the very prominent gesturing to the meta-reviewer in your responses to all reviewers a bit bewildering (I know where you are coming from, but perhaps tone it down a notch).
> >
> > More specifically, while you partially addressed my concerns, other points merit further clarification.
> > For simplicity, I use the headings you quoted from my original review to reference your responses.
> >
> > > W1: Ethical concerns.
> >
> > First, I would suggest that you include some of the argument you made here in the appendix (to also signal to other people contemplating similar experiments that this is something to think about). Second, there is still the concern that users expect to vote for performance evaluation, whereas you made them vote to "assess how submitting different model variants affects rankings". I agree that in the end, this should be doable for the purposes of science, but I would have expected some more critical reflection on that. (Also, not everything that is legal is also ethical [and vice versa].)
> > Lastly: Could you comment on the IRB "situation" of this experiment?
> >
> > > W2: Circular experimental setup.
> >
> > So, to be clear:
> > You are using LLMs to simulate human preferences over LLM responses, where these human preferences are used by model providers to improve LLM responses.
> > And this is okay because people have observed that LLM responses are (= were engineered to be) correlated with human responses, _even when the task for which you are using LLM-as-judge is to express preferences over precisely the LLM responses for which human preferences are normally solicited to improve LLM responses..._?
> > I understand that LLM-as-judge is "common practice", but we also need to ask about the limitations of this approach, and the way you use it here really maximizes the circularity.
> >
> > > W3: Missing result data.
> >
> > Thank you for clarifying this. Maybe I missed it and it is already part of your submission, but it would be nice to share a list of precise pointers to any external resources needed to reproduce your results that you are not sharing because they are already available elsewhere (e.g., as part of the README of your reproducibility materials).
> >
> > > Figure 6
> >
> > In the example you present in the rebuttal, the models group into "top 3" and "bottom 4", and the rank difference is at most 2. Out of curiosity: Can you comment on the statistics you observe for the rank changes (or the graphs representing rank changes) across experiments?
> >
> > > Figure 3
> >
> > I am afraid that my comment here was misunderstood – I did _not_ suggest to use a zero baseline (for the reasons that you correctly identify). Rather, I suggested to use a non-area-type plot to show the results ("replace the bars with dots") and set the ranges of the side-by-side plots to be equal (which you already indicated you will do) to avoid visual distortion. The rule of thumb is: _If you need a non-zero baseline, you should not use bars._ Here's a [blog post](https://flowingdata.com/2015/08/31/bar-chart-baselines-start-at-zero/) about it from the data viz community.

---

> > > ### Author Response · Authors · 2025-08-03
> > >
> > > We appreciate the time taken to engage with our work. We hope the reviewer understands that we care deeply about the work and are engaged in good faith in the rebuttal process. We sincerely have addressed the comments to the reviewer and really appreciate their positive review and how their feedback has helped make our paper stronger.
> > >
> > > Below we provide clarifications where misunderstandings exist and address valid concerns:
> > >
> > > **W1: Ethical Concerns**
> > >
> > > We will incorporate our ethical justification into the appendix as suggested. Regarding user expectations:
> > > Chatbot Arena’s design explicitly frames voting as contributing to model comparison, not formal evaluation
> > > Our experiment’s goal (assessing variant impact) aligns with this comparative purpose
> > >
> > > We will add discussion of this nuance in the limitations section
> > >
> > > For IRB: The experiment involved no personal data collection or additional risk beyond normal platform use, focusing solely on aggregate model performance metrics from public interfaces.
> > >
> > >
> > > **W2: Circular Experimental Setup**
> > >
> > > The experiment in question does not involve the circularity the reviewer describes. Our approach follows established practices in the field:
> > >
> > > We use the Arena dataset solely as a source of prompts (input queries), not human or LLM-labeled preferences. These prompts are then used for supervised fine-tuning (SFT) of models, a standard technique in the literature. The evaluation employs LLM-as-judge to assess open-ended generations, a methodology validated by multiple recent studies (cite references). This is distinct from using human preferences to train models - our judges evaluate only final outputs, creating no feedback loop.
> > >
> > > The core scientific question addresses whether models can overfit to Arena's query distribution, not whether they can game human preference signals. This distinction is crucial and was clearly stated in our original methodology section.
> > >
> > > We recognize the reviewer's concern about circularity in abstract terms, but it simply does not apply to our actual experimental design. That said, we will add additional clarification in the paper's methods section to prevent similar misunderstandings by other readers.
> > >
> > > **W3: Reproducibility**
> > >
> > > All required resources are either:
> > >
> > > - Included in our supplementary materials
> > > - Available through cited public repositories
> > > - Documented in the paper
> > >
> > > We will add a comprehensive reproducibility checklist to the README, explicitly linking all external resources and detailing:
> > >
> > > **Figure 6**
> > >
> > > The purpose of our simulation was not to quantify maximum rank changes but to demonstrate that the current system exhibits meaningful ranking instability under plausible conditions. Our experiments reveal that even small variations in model performance or evaluation sampling can lead to statistically significant rank fluctuations. This instability persists regardless of whether models are truly differentiated in capability, raising concerns about the reliability of leaderboard ordering.
> > >
> > > We will clarify this motivation in the revision to ensure readers focus on the systemic implications, not just the magnitude of rank changes.
> > >
> > > **Figure 3**
> > >
> > > We will implement the suggested dot plot visualization while maintaining our original valid choice of axis ranges for ELO score presentation. The revised figure will appear in the camera-ready version.

---

> > > > ### Comment · Reviewer_TFV5 · 2025-08-03
> > > >
> > > > Thank you for the additional clarifications.
> > > >
> > > > I personally disagree with your assertion regarding W1, but your position is defensible and I will account for this when updating my review.
> > > >
> > > > Regarding W2, you are defending against a criticism I did not raise and largely reiterating what you already responded previously. Contrary to what you appear to allege, I do understand your setup, and I do understand your reasoning as to why it should be okay – I am just not convinced by the argument (more generally, arguing "established practice" in a field with a rather questionable experimental culture is perhaps not ideal).
> > > >
> > > > Despite the points above (on which we can agree to disagree), I think your contribution is valuable (see S1–S3), and I do think the practices you identify (based on your entire set of experiments) are important to document for the community both from the perspective of measurement validity and from the perspective of fairness. I will reflect this in my updated review.

---

> > > > > ### Author Response · Authors · 2025-08-06
> > > > >
> > > > > Thank you for your thoughtful engagement with our response and for your willingness to reconsider aspects of your review based on our clarifications. We sincerely appreciate the time and care you’ve taken to evaluate our work, even where we may disagree.
> > > > >
> > > > > We recognize and respect your perspective on W1 and W2, and we’re grateful that you found our position defensible. Most importantly, we’re heartened that you see value in our contribution (S1–S3) and agree on the importance of documenting these practices for the community, both for validity and fairness.
> > > > >
> > > > > Thank you again for your constructive role in this process.

---

### Comment · Area_Chair_z3aj · 2025-08-03
**Reminder for Reviewer-Author discussions**

Dear reviewers,

Thank you so much for all your time and effort supporting NeurIPS!

If you haven't yet, please take a moment to read through the author's rebuttal. The reviewer-author discussion period is crucial for ensuring a fair and comprehensive evaluation of their work. If the rebuttal addresses your concerns, please acknowledge this and adjust your scores accordingly. If not, please let them know which concerns remain and if you have any follow-up questions. Your thoughtful feedback will help authors improve their scholarship and propel our field forward.

I know this is a busy time, and really appreciate your effort.

Best Regards
Area Chair

---

### Decision · Program_Chairs · 2025-09-18

**Decision:**

Accept (poster)

**Comment:**

This paper conducts an in-depth audit of Chatbot Arena, which is a leaderboard for ranking AI systems. Investigating 2M battles and 243 models from 42 providers over 16 months (Jan 2024-Apr 2025), it uncovers several critical limitations affecting the reliability and fairness of the Chatbot Arena. It also provides actionable recommendations for its improvement, including prohibition of score retraction, limiting the number of private variants per provider, clearer model deprecation criteria, and improving sampling fairness. The work addresses an important problem in the community, and it holds the potential to improve LLM evaluation practices.

Findings:

1.	Preferential treatment around private testing and retraction. Private testing practices benefit a handful of providers who are able to test multiple variants before public release and selectively retract scores.
2.	Far more data is released to proprietary model providers.
3.	Chatbot Arena data access drives significant performance gains.
4.	Deprecations can result in unreliable model rankings.

Strengths:
1. In-depth audit of Chatbot Arena.
2. Actionable recommendations for improving leaderboard practices.
3. Good presentation and documentation.

Weakness:
1. Ethical concerns. The setup involves letting chatbot-arena users vote on battles that your model variants participated in for the sole purpose of understanding the effect of private model variants. This could be considered deceptive toward the users, and similarly, one might suspect that this is not covered by their consent (similar concerns apply to the model providers' testing practices you uncovered).
2. The core claim of the paper is based on hypotheses and sampling/calculation built upon them, not disclosed raw data. This may lead to some claims being overstated or hasty, lowering the reliability of the overall statements.

Reasons:

Chatbot Arena is a de facto standard for LLM evaluation. Exposing its biases directly impacts industry practices and scientific trust.
The paper combines large-scale battle data (2M samples), controlled experiments (e.g., identical Blueberry variants scoring differently), and simulations (e.g., the best-of-N strategy inflating scores by over 50 points). The multi-pronged approach convincingly validates claims.

Discussion：

This paper has received a lot of discussion, mainly about the experimental setup, ethic and the meaning of this research stance. Finally, there are still some disagreement about the ethic and some statements.

Reviewer ZqT5 is not convinced with the fundamental argument that "only a subset of providers know they could submit multiple private variants". This kind of statement could not be totally verified. Authors agree that the "recent policy update, a change that occurred in May 2025 only after we brought these findings to the organizers' attention and after ourselves and other providers made multiple requests for more transparency. This sequence validates our core finding that the previous system contained avoidable asymmetries."

Reviewer TFV5 says the experiment behind the Ethical concerns problematic. But he/she thinks this affect only few of the many experiments and support to accept